# Effect of Crosslinking Using Heat on the Physicochemical Features of Bsa–Capsaicin Nanoparticles

**DOI:** 10.3390/pharmaceutics17101306

**Published:** 2025-10-08

**Authors:** Lino Sánchez-Segura, Silvio Zaina, Angela F. Kú-González, José Alfredo Guzmán-López, Laura E. Zavala-García, Mercedes G. López

**Affiliations:** 1Departamento de Ingeniería Genética, Centro de Investigación y de Estudios Avanzados del Instituto Politécnico Nacional—Unidad Irapuato, Guanajuato 36824, Guanajuato, Mexico; 2Department of Medical Sciences, Division of Health Sciences—Leon Campus, University of Guanajuato, Leon 37320, Guanajuato, Mexico; 3Unidad de Biología Integrativa, Centro de Investigación Científica de Yucatán, Mérida 97200, Yucatán, Mexico; 4Unidad de Genómica Avanzada, Centro de Investigación y de Estudios Avanzados del Instituto Politécnico Nacional, Irapuato 36821, Guanajuato, Mexico; 5Departamento de Biotecnología y Bioquímica, Centro de Investigación y de Estudios Avanzados del Instituto Politécnico Nacional—Unidad Irapuato, Guanajuato 36824, Guanajuato, Mexico

**Keywords:** crosslinking, heat, bovine serum albumin, capsaicin, steatosis, nanoparticles

## Abstract

**Background/Objectives**: The synthesis of protein nanoparticles (NPs) using the coacervation method is influenced by critical parameters. The use of glutaraldehyde limits the pharmacological applications of NPs in humans due to the potential toxicity of residual aldehydes that remain after the purification of the nanoparticles. The aim was to assess heat effect as a crosslinking agent for the synthesis of bovine serum albumin (BSA)–capsaicin nanoparticles and its effect on the physicochemical characteristics of nanoparticles. **Results**: The initial concentrations of BSA and capsaicin in the formulation were directly correlated with the amount of BSA that was transformed into nanoparticles and the loaded capsaicin (*r* = 0.97, *p* = 0.0003 and *r* = 0.95, *p* = 0.0003), respectively. Furthermore, the morphometric parameters of nanoparticles were affected by the increase in capsaicin concentration, but not by temperature. The nanoparticles increased in dimensions and showed a loss of shape due to coalescence between nanoparticles. The ζ-potential decreased with the increase in the concentration of capsaicin added. This effect compromised the stability of the nanoparticles; on the other hand, molecular interactions were observed between hydrophobic residues of phenylalanine and tyrosine in BSA and the hydrophobic moiety of capsaicin. At the same time, BSA nanoparticles showed a potential for disassembling and delivering the payload capsaicin, which caused an antisteatotic effect in the liver of a murine model. **Conclusions**: heat (70 °C) can replace crosslinking agents, such as glutaraldehyde. This property is particularly useful when an aldehyde-free synthesis of BSA nanoparticles is needed.

## 1. Introduction

The interest in nanoencapsulation of bioactive compounds has increased in recent years. Among the most studied bioactive plant metabolites is capsaicin [(E)-N-(4-hydroxy-3-methoxyphenyl)methyl-8-methylnon-6-enamide] due to its biological properties, such as antioxidant, anti-inflammatory, analgesic, anti-tumour, regulator of lipidaemia, cardio-protective, anti-calculus, thermogenic, regulator of circadian cycle, and promoter of healthy gastrointestinal function [1,2,3]. Initial studies focused on the development of nanoemulsions and polysaccharide nanocapsules with capsaicin [4,5,6]. An additional strategy involved encapsulation with a protein biopolymer using bovine serum albumin (BSA) to improve solubility in water and provide high stability when exposed to environmental conditions and tolerance to high temperatures and photo-oxidation [7,8,9]. The albumin biopolymer and its applications, such as in nanoparticles, have played an important role in elucidating several mechanisms, including molecular assembly, physicochemical features, stabilisation, mechanism of internalisation in cells, drug delivery, recycling, and elimination [10,11,12]. Historically, glutaraldehyde has been used as a crosslinking agent due to its strong and rapid intermolecular bonds at low concentrations (1%). It is soluble in water and miscible in several solvents such as ethanol, acetone, isopropanol, toluene, hexane, ethyl acetate, methylene chloride, and ether [13]. Protein modification using glutaraldehyde is irreversible and resistant to urea oxidation and semicarbazide, and extreme changes in pH, ionic strength, and high temperatures, and does not affect the crystal structure of some enzymes [14]. It has been postulated that glutaraldehyde is a better crosslinking agent than other available molecules, such as formaldehyde [15]. Among the advantages of glutaraldehyde is the modest protein denaturation, which is a desired feature if the original activity of the NP protein needs to be preserved. Glutaraldehyde use has some disadvantages; for example, the NPs exhibit coalescence and aggregation at pH > 8, while the crosslinking activity decreases at low pH [15]. Additionally, some bioactive compounds, such as nutrients, phytochemicals, nutraceuticals, and secondary metabolites, react with some functional groups in proteins under oxidising conditions, generating natural crosslinking [16]. Furthermore, glutaraldehyde has been shown to have several adverse effects on the physicochemical properties of BSA NPs, including decreasing the electrophoretic mobility [17]. An additional important disadvantage of glutaraldehyde is the potential toxicity of residual aldehydes even after NP purification, which limits clinical applications [11,18,19].

In pharmaceutics, the desolvation–pH–coacervation method is extensively applied for the micro- and nanoencapsulation of drugs with biomedical polymers such as albumin [20,21]. The synthesis of nanoparticles via the coacervation method has several critical stages that are affected by initial protein concentration, temperature, pH, glutaraldehyde concentration, agitation speed, and organic solvent addition rate [22]. Glutaraldehyde has several adverse effects on the physicochemical properties of BSA nanoparticles, including decreasing the electrophoretic mobility of NPs [17]. Therefore, in the case of BSA nanoparticle synthesis and the loading of inorganic enzyme-like catalytic activity materials, temperature control must be considered as an alternative option to primary crosslinking, leaving the possibility of glutaraldehyde as a crosslinking agent to functionalise the BSA nanoparticles [21].

The aforementioned limitations of glutaraldehyde have prompted efforts to identify suitable alternatives. Crosslinking using heat is a promising alternative for achieving primary crosslinking in the synthesis of and payload incorporation into BSA NPs [21]. However, the use of heat as a crosslinking agent to obtain BSA NPs loaded with plant metabolites such as capsaicin has not been investigated. Here, we assessed the feasibility of synthesising capsaicin-loaded BSA NPs via heat crosslinking. Since the synthesis of NPs via the coacervation method is affected by several critical parameters such as initial protein concentration, temperature, pH, crosslinker concentration, agitation speed, and organic solvent addition rate, we measured the efficiency of capsaicin-loaded BSA NP synthesis across a variety of conditions [22].

## 2. Materials and Methods

### 2.1. Chemicals

The biological reagents used in this study were BSA (lyophilised powder, 66 kD) (Equitech-Bio, Kerrville, TX, USA) and pharmaceutical-grade capsaicin (≥99% purity) from *Capsicum* spp. (Handim Chemical Co., Ltd., Shanghai, China). The chemical reagents were analytical-grade sodium chloride (Merck, Darmstadt, Germany), acetonitrile (≥99.9% purity) (Sigma-Aldrich, St. Louis, MS, USA), and absolute ethanol (≥99.8% purity) (Merck, Darmstadt, Germany).

### 2.2. Synthesis of BSA NPs Loaded with Capsaicin

To control the temperature or heat during the synthesis of NPs, a single glass beaker was adapted to a reactor to achieve a reproducible process and low-cost operation. The main technical difficulties were establishing heat transfer via the heat plate, monitoring the temperature in the reaction chamber, and administering the ethanol–capsaicin solution. This process was carried out in a semi-closed system to reduce the evaporation of ethanol. We designed a reactor with a 50 mL capacity that allowed for the introduction of a propeller for agitation, administration of solutions, and monitoring of temperature while remaining semi-closed during coacervation and the crosslinking process.

NPs were prepared using a desolvation technique modified from Langer et al. [20], Sánchez-Segura et al. [8], and Khramtsov et al. [21]. Briefly, 120 mg of BSA powder was dissolved in 4 mL of sterile deionised water and adjusted to pH 10 with pure sodium hydroxide powder. The solution was maintained under agitation at 1000 rpm using a stirrer (Eurostar 20, IKA, Wilmington, NC, USA) for 15 min at room temperature. Coacervation was induced by the addition of 16 mL of ethanol–capsaicin solution at 0, 65, 130, 195, and 260 µg per mg of BSA for each treatment. The addition rate was 1.5 mL/min at 1000 rpm. Crosslinking was conducted by heating the nanoparticle solution at 70 ± 1 °C for two hours under agitation at 1000 rpm under dark conditions. The NPs were washed via three cycles of centrifugation at 12,485× *g* for 40 min at room temperature (MC-12V, DuPont, Newtown, CT, USA), and the pellet was dispersed in sterile deionised water in every cycle. The pellet was resuspended by sonication for 10 min using a digital ultrasonic machine (PS-20A, VEVOR, Wuhan, China). The purified NPs were stored at 4 °C.

### 2.3. Quantification of BSA Transformed into NPs and Encapsulated Capsaicin

Capsaicin molecules encapsulated in NPs were extracted via recovery using acetonitrile, as reported by Sganzerla et al. [23] and modified by Sánchez-Segura et al. [8] and Sánchez-Arreguin et al. [9]. Volumes of 500 μL of nanoparticles were washed with deionised water through three cycles of centrifugation at 12,485× *g* for 40 min at room temperature; the supernatant was discarded in the final step. The pellets were lyophilised (HETO MAXY dry, LYO, Lillerød, Denmark), and the dried NPs were broken by adding 250 mL acetonitrile. The samples were homogenised for 1 min (Super Mixer, LAB LINE Melrose Park, IL, USA) and sonicated for 15 min at 30 °C. The samples were then centrifuged at 12,485× *g* for 30 min (MC-12V, DuPont, Newtown, CT, USA). Finally, the supernatants were filtered through an acrodisc with a 0.22 µm pore size (Sartorius Lab Instruments GmbH & Co. KG, Goettingen, Germany) and directly deposited in a HPLC vial. The samples were stored at −20 °C. Denaturalised protein was incubated at 30 °C for one hour to evaporate residual acetonitrile. Sample weights were measured using an analytical balance (Pioneer, Ohaus Analytical Plus, Shanghai, China). BSA NP yields were calculated using equations described by Bhaleka et al. [24]. The estimation of BSA transformed into NPs was adjusted to the total volume recovered from the nanoparticle coacervation process. The proposed Equation (1) was computed for each experiment as follows:(1)$$BSA\;NP\;yield\left(\%\right)=\frac{BSA\;in\;NP\left(mg\right)}{initial\;BSA\left(mg\right)}\times \;100$$

Capsaicin was quantified using HPLC-UFLC (SPD-20A, Shimadzu, Camby, OR, USA). Capsaicin separation was achieved using a Shimadzu Shim-pack GIST C18 column with a 100 Å pore size (3 μm, 4.6 mm × 50 mm) (Part number: 227-30011-03, Shimadzu) and reverse phase consisting of water (A) and acetonitrile (B) [A:B (40:60, *v*/*v*)] in isocratic mode at a flow rate of 1.0 mL/min. The absorbance of the eluted material was measured at 280 nm, and ultraviolet (UV) spectra were recorded in the range of 220–350 nm at an acquisition rate of 1.25 scan/s. Calibration curves were generated using different concentrations (0, 0.125, 0.25, 0.5, 1.0, and 2.0 mg/mL) of a capsaicin standard. The curve and samples were prepared by injecting 10 μL of the solution in triplicate. The total quantity of capsaicin loaded into NPs was estimated by adjusting the total volume recovered from the NP coacervation process.

### 2.4. Measurement of Particle Size, PDI, and ζ-Potential

The hydrodynamic particle size diameter, polydispersity index (PDI), and ζ-potential of purified NPs were determined through dynamic light scattering (Zetasizer Nanoanalyser ZS-90P, Malvern Instruments, Malvern, UK). To measure size, the samples were diluted 1:20 with deionised water and measured at a scattering angle of 90° and a temperature of 25 °C in a particle size analytical sample cell (DTS0012, Malvern Instruments). Data were automatically evaluated using the Smoluchowski equation, in which a particle size of ≈100 nm is much larger than the Debye length of ≈1 nm [25]. To determine the ζ-potential, the sample was charged in a zeta potential sample cell (DTS1070, Malvern Instruments). Data from zeta potential graphs were analysed using SigmaPlot 12 (Systat Software, Inc., Sausalito, CA, USA).

### 2.5. Fourier Transform Infrared (FTIR) Spectroscopy

The molecular interactions between BSA and capsaicin were examined using FTIR. The spectra were recorded using an FTIR spectrometer iS50 Nicolet (Thermo Scientific, Waltham, MA, USA) equipped with an attenuated total reflectance (ATR) diamond array. Samples were powdered by freeze-drying and scanned at a wavenumber range of 400–4000 cm^−1^ in transmittance mode [%T]. Capsaicin standard and BSA samples were homogenised and analysed directly, whereas NP samples were washed with deionised water through three cycles of centrifugation at 12,485× *g* for 30 min at room temperature. The supernatant was discarded after the final step. The pellet was lyophilised (HETO MAXY dry, LYO, Lillerød, Denmark) for three hours and subsequently homogenised and analysed. Sixty-four scans were recorded with a nominal resolution of 4 cm^−1^ in transmittance mode [%T]. Single-beam spectra of the samples were collected against an air background. Three replicates of each sample were averaged to obtain one spectrum. Spectral data were decoded using the spectroscopy software SpectraGryph version 1.2 (Dr. Friedrich Menges Software-Entwicklung, Oberstdorf, Germany). The spectral graphs were analysed using SigmaPlot 12 (Systat Software, Inc.).

### 2.6. Raman Analysis

The samples were analysed using a Thermo Scientific DXR3xi Raman Imaging Microscope (Thermo Fisher Scientific, Waltham, MA, USA). The equipment was used in normal Raman analysis mode. The excitation wavelength was 532 nm, with a laser power of 20 mW and an exposure time of 1 s. The scan was measured at 100–3500 cm^−1^ wavenumber range. For the analysis of capsaicin and BSA standards, the powder was homogenised and mounted on glass slides, covered with high-performance Zeiss cover glasses (D = 0.17 ± 0.005 mm, refractive index = 1.5255 ± 0.0015, Abbe number = 56 ± 2) and analysed directly. The BSA–capsaicin NPs were washed and lyophilised as mentioned in the FTIR analysis. The samples were observed and analysed with objective 50×/1.4, NA ∞−0.17, Olympus Plan FLUAR. Three replicates of each sample were averaged to obtain one spectrum. Spectral data were decoded using the spectroscopy software SpectraGryph version 1.2 (Dr. Friedrich Menges Software-Entwicklung, Oberstdorf, Germany). The spectral graphs were analysed using SigmaPlot 12 (Systat Software, Inc.).

### 2.7. Transmission Electron Microscopy (TEM) and Morphometric Analysis of Nanoparticles

NP morphology was examined using TEM (Morgagni M-268, Philips/FEI, Brno, Czech Republic). NP samples (7 μL) were placed on 200 mesh formvar/carbon-coated copper grids (Ted Pella Inc., Redding, CA, USA) and incubated for 7 min. Samples were contrasted with 2.0% uranyl acetate (Electron Microscopy Science Inc., Hatfield, PA, USA) and incubated for 10 min. The TEM operating conditions for all experiments were 80 kV high voltage (EHT) from 8900× to 9000× for low magnification (shadow magnification) and from 56,000× to 180,000× high magnification at a low vacuum pressure of 5 × 10^−3^ Pa (5 × 10^−5^ Torr). Micrographs were captured in tagged image file (.tif) format with 1376 × 1032 pixels in grey scale. In this format, 0 was assigned to black and 255 to white in the greyscale.

The NP images obtained from each treatment were cropped and resized to a new image of 1376 × 1032 pixels in .tif format with 8-bit compression on a white background. The resolution of the images was 2.1 pixels/nm. The morphometric descriptor was calculated using the “shape_descriptor1u” plugin for ImageJ v.1.49p software (National Institutes of Health, Bethesda, MD, USA). The parameters of the particles were area (*A*), effective diameter (*Ed*), aspect ratio (*Ar*), and the shape factor (*Sf*). *A* was calculated as the sum of pixels within a selection or a measured region of interest, optionally converted to calibrated units if the image had a defined scale. *Ed* calculates the diameter of a fictitious circular object that has the same area as the object being measured, according to Syverud et al. [26] (Equation (2)). *Ar* was calculated according to Parakhonskiy et al. [27] and describes the relationship between the width and length of the particles. *Ar* is a suitable descriptor of ellipticity, e.g., for “flat” particles, *Ar* < 1, for circular particles, *Ar* = 1, and for elliptical particles, *Ar* > 1 (Equation (3)). *Sf* or circularity is based on the projected area of the particle and the overall perimeter of the projection, as described by Bouwman et al. [28] (Equation (4)).(2)$$Ed=\sqrt[{2}]{\frac{A}{\pi }}$$(3)$$Ar=\frac{Majoraxis}{Minoraxis}$$(4)$$Sf=\frac{4\;.\;\;\pi \;.\;\;A}{P^{2}}$$

### 2.8. Sodium Dodecyl Sulphate–Polyacrylamide Gel Electrophoresis (SDS-PAGE)

To distinguish the disassembled BSA molecules from BSA and BSA–capsaicin NPs, sodium dodecyl sulphate polyacrylamide gel electrophoresis (SDS-PAGE) was performed using the Tris–glycine buffer system of Laemmli [29] modified by Carriles et al. [10]. To estimate the protein molecular weight (MW), we used the EZ-Run pre-stained *Rec* protein ladder with bands in the range of 10–170 kDa (Fisher Scientific, Waltham, MA, USA). BSA, BSA–capsaicin NPs, and BSA solution (protein content 4 mg/mL) were boiled in SDS sample buffer (50 mM Tris–HCl, pH 6.8, 2% SDS, 0.1% Coomassie Brilliant Blue (CBB), 10% glycerol) containing 100 mM β-mercaptoethanol for 5 min. Twenty microliters of sample aliquots were loaded onto a 4% stacking gel and separated by electrophoresis in a 12% resolving gel. The samples were resolved at a constant voltage of 60 V for stacking and 100 V for resolving. The gels were washed three times with deionised water for 5 min. The gels were stained with CBB staining solution (0.025% Coomassie dye, G-250 in 10% acetic acid). The gels were then washed again with deionised water and incubated overnight with agitation and clarified to visualise polypeptide bands. Gel images were captured using the Gel Doc^TM^ XR (Documentation Systems Bio-Rad, Hercules, CA, USA) at 600 dpi resolution in .tif file format with a resolution of 1580  ×  1489 pixels in RGB (Red, Green, and Blue).

### 2.9. Administration of BSA–Capsaicin NPs to a Murine Model

To probe the biocompatibility of BSA–capsaicin nanoparticles, twelve Swiss Webster mice (BALB/c) were injected with nanoparticles diluted with 10 mM NaCl (control). All animals were provided ad libitum access to food and water. The nanoparticles administered comprised 5 doses administered over 15-day intervals. The route of administration was subcutaneous, with each dose consisting of 40 μL of NPs containing 0, 65, and 260 µg of capsaicin (empty, low, and high capsaicin loading ratio resuspended in 10 mM NaCl), and a control of 40 μL of 10 mM NaCl. The mice were euthanised and their livers collected for histopathological analysis and multiphoton microscopy. All experiments were approved by the Institutional Care and Use of Laboratory Animals Committee (CICUAL), protocol number 0342-22.

### 2.10. Histopathology of Mouse Liver

Livers were cut into four pieces, 3 and 4 mm thick, and fixed in 4% paraformaldehyde solution dissolved in 0.16 M phosphate buffer (PB) for 12 h at 4 °C. Subsequently, the tissues were washed three times with PB and then dehydrated in a series of seven graded ethanol concentrations (40–100%) for 30 min at room temperature. Samples were incubated overnight in 100% ethanol and transferred to a graded ethanol–xylene solution at ratios of 2:1, 1:1, and 1:2 for 1 h in each solution. These materials were immersed in a pure xylene solution for 3 h and finally placed in pure Paraplast^®^ at 60 °C in the form of blocks, which were then used to obtain transverse sections. Thick sections (10 μm) were prepared using a rotatory microtome (LKB, Bromma, Sweden). The samples were then rehydrated in a graded ethanol series and stained with haematoxylin–eosin (Electron Microscopy Science, Hatfield, PA, USA) and embedded in Entellan^®^ for observation under an optical microscope (Olympus BX50, Tokyo, Japan).

### 2.11. Multiphoton/Confocal Microscopy

Liver samples were fixed with 4% paraformaldehyde solution dissolved in 0.16 M phosphate buffer (PB) for 12 h at 4 °C. Subsequently, the tissues were washed three times with PB and then embedded in Leica tissue freezing medium (Leica Biosystems, Milton Keynes, UK) and cut into sections of 20 µm thickness. The sections were washed with PB, stained with 3 µL of Nile red from a stock solution of acetone–Nile red [500 µg/mL] dissolved in a fresh solution of 75% glycerol (Sigma-Adrich, St. Louis, MO, USA) and incubated for 30 min at 4 °C. The second staining consisted of a mixture of 1.5 µL DAPI (Sigma-Adrich, St. Louis, MO, USA) and two drops of ActinGreenTM 488 (Thermo Scientific, Waltham, MA, USA, USA) to stain the actin in the cytoskeleton. Finally, the sections were washed with distilled water and embedded in FluoroshieldTM mounting medium (Sigma-Adrich, St. Louis, MO, USA). Observations were performed using a multiphoton microscope system (LSM 880-NLO, Zeiss, Oberkochen, Germany) coupled to an infrared laser Ti/Sapphire (Chameleon vision II, COHERENT, Santa Clara, CA, USA). The operating conditions in all experiments were as follows: argon laser at 488 nm at 1.8% power and 543 nm at 1.2% power, a Chameleon laser at 760 nm at 1.5% power, and an open pinhole. All samples were observed using an immersion objective (40X/1.3, NA ∞−0.17, Zeiss Plan NEOFLUAR). Images were acquired by separation of the emission in three channels: DAPI (411–459 nm) for nucleus, Alexa 488 (476–511) for Actin, and Nile red (501–617 nm) for lipid droplets. All micrographs were captured in CZI format at 2048 × 2048 pixels and RGB.

## 3. Results and Discussion

### 3.1. Yield of BSA Transformed to NPs and Quantification of Loaded Capsaicin

The quantities of native BSA transformed into nanoparticles increased with initial capsaicin concentration (treatments ranging from 0 to 195 µg of capsaicin loading ratio), showing a linear trend (*r* = 0.97, *p* = 0.0003) (Figure 1). However, at the 260 µg capsaicin loading ratio, the BSA transformed into nanoparticles showed an adverse effect (Figure 1, red circle). The native BSA showed the capacity to change from isoform A to isoform B or N following a change in the basic pH [30]. Based on these facts, we propose the hypothesis that the decrease in native BSA transformed into nanoparticles was probably caused by the reversibility of BSA isoform A to N. The change in pH was probably caused by a higher concentration of capsaicin. Kogure et al. [31] found that capsaicin effectively scavenged OH radicals in water. The functional OH was dissociated from NaOH used to adjust the pH to 10. Additionally, the impact of the loaded drug was dependent on the chemical characteristics of the molecule. However, the effect of drug concentration was observed in several studies of BSA nanoparticles, where the molecules loaded were simple salts, such as NaCl, or complex hydrophobic drugs, such as paclitaxel (PTX), due to alterations in the electrostatic properties of BSA or saturation of the hydrophobic sites of the BSA by the drug [32,33]. The yield of native BSA transformed into NPs decreased by approximately 11.6% relative to the loading ratio of 195 µg of capsaicin; hence, the limit of the capacity to entrap capsaicin was found to be 195 µg of capsaicin, which is 1.8 times more than the maximum quantities reported by Sánchez-Arreguin et al. [9], who used crosslinking with glutaraldehyde. Therefore, heat-based crosslinking had two advantages: it allowed the display of the hydrophobic sites in albumin and was an effective way of promoting a peptide bond between BSA molecules; however, this process had an adverse effect: above the melting point of BSA (*T*_M_ = 62 °C), the relatively hydrophobic conformation is permanent, and the intermolecular hydrophobic interactions increase [34,35]. Therefore, at a loading ratio of 260 µg of capsaicin, the hydrophobic activity of capsaicin probably increases the coalescence between particles, subsequently forming small aggregates and disassembling the nanoparticles. Hence, the initial concentration of BSA can be adjusted to allow higher concentrations of capsaicin to be entrapped. The yield was calculated as described by Sánchez-Segura et al. [8] for loading ratios of 0, 65, 130, 195, and 260 µg of capsaicin, obtaining average values of 58.4 ± 4.4%, 61.7 ± 6.8%, 69.9 ± 5.0%, 77.8 ± 9.1%, and 66.2 ± 8.0%, respectively. Temperature was excluded as a cause of BSA agglutination, as lower concentrations of capsaicin did not exhibit any effect.

On the other hand, quantification of the capsaicin loaded into the nanoparticles showed a linear correlation (*r* = 0.95, *p* = 0.0003) with initial capsaicin concentration (Figure 2). However, treatments at higher concentrations did not have an adverse effect, as observed for the BSA transformed into NPs. This effect may be attributed to the fact that the BSA molecules are more sensitive to exposure to constant temperatures (70 °C for 2 h) than capsaicin; based on the following rationale, capsaicin is not expected to undergo chemical decomposition at the crosslinking temperature. Some studies found that the thermal decomposition of capsaicin occurred in three stages. The first stage was reached at a heating rate of 10 °C·min^−1^, from 23 to 265 °C, and only corresponds to the evaporation of moisture in the capsaicin powder, causing slight damage [36]. Chemical decomposition occurred within the temperature range of 265–425 °C [36,37,38]; therefore, the crosslinking temperature does not affect the chemical stability of the capsaicin molecule.

### 3.2. NP Morphology

To evaluate the effective temperature of crosslinking and the optimisation of the formulation, void and capsaicin-loaded BSA NPs were evaluated using TEM, as described in previous reports [8,9]. Void nanoparticles (loading ratio of 0 µg of capsaicin) had a smaller size compared with the capsaicin-loaded counterparts. The NP aspect was approximately spherical, with some flattening at the poles, causing some irregular aspect (Figure 3a). In the panoramic image, the nanoparticles showed a homogeneous size distribution, and aggregation of nanoparticles was not observed (Figure 3f). The size appeared to increase at a loading ratio of 65 µg of capsaicin, and the shapes of the NPs were more circular, with irregular borders and the presence of small particles that were probably not condensed into a nanostructure (Figure 3b). The panoramic image showed an increase in the dispersion of sizes (Figure 3g). The NPs isolated at a loading ratio of 130 µg of capsaicin were circular in shape (2D image in TEM micrographs), probably corresponding to the spherical shape of the nanoparticles in the 3D plane. The size of the NPs increased, and the core (osmiophilic affinity) encompassed the complete internal space, probably indicating that the entrapped capsaicin had reached its maximum capacity (Figure 3c). The panoramic image showed homogeneity in the size dispersion of NPs, and no aggregated particles were observed (Figure 3h). In treatments with the high concentration of capsaicin (195 µg loading ratio), some nanoparticles showed the formation of condensed bilobulated particles composed of two particles—one large particle and one small particle (Figure 3d). This effect was probably due to increased surface electrical charges of the nanoparticles, causing attraction of the nanoparticles with positive charges (small particles). The panoramic image showed heterogeneity in particle sizes, as well as several aggregates composed of 2 or 3 particles (Figure 3i). A loading ratio of 260 µg of capsaicin showed aggregation of some nanoparticles; however, the isolated nanoparticles were predominantly of large sizes (Figure 3e). Moreover, the panoramic image revealed the formation of some aggregates and isolated particles with large sizes (Figure 3j). In this treatment, residual albumin that was not nanostructured was observed; this was probably due to the denaturation of BSA due to the effect of temperature and capsaicin concentration, which caused BSA precipitation and a decrease in the nanoparticle yield.

### 3.3. NP Morphometric Parameters

The change from traditional glutaraldehyde crosslinking agents to heat, and the increased concentration of capsaicin in the formulation, affected morphometric parameters, as reported by Sánchez-Segura et al. [8], Sánchez-Arreguin et al. [9], and Carriles et al. [10], were an increase in size and aggregation with increasing loading ratio of capsaicin was observed. However, under the conditions used in this work, the loading ratio of 0 µg of capsaicin showed a lower area of isolated nanoparticles (4245.4 ± 660.2 nm); this morphometric characteristic of empty NPs is illustrated in the box plot (Figure 4). The *A* of the NP increased in response to the capsaicin concentration. The area had the following average values: 5639.4 ± 821.5, 7813.9 ± 1028.0, 10,426.0 ± 1780.4, and 15,208.3 ± 2406.3 at loading ratios of 65, 130, 195, and 260 µg of capsaicin, respectively. This trend is illustrated in the box plot (Figure 4). The area of the NP increased 3.5 times with respect to the empty nanoparticles.

The diameter of nanoparticles is usually reported in nanometres; however, the diameter is a confidence parameter under two conditions: the population of nanoparticles is isolated, and they have a spherical shape. In this study, the empty nanoparticles (0 µg of capsaicin) satisfy both conditions; thus, they can be used to compare the effect of the loaded capsaicin and the effect of crosslinking. The average effective diameter (*Ed*) was 72.7 ± 5.5, 84.8 ± 5.7, 101.6 ± 7.4, 120 ± 17.6, and 151.1 ± 17.8 nm at loading ratios of 0, 65, 130, 195, and 260 µg of capsaicin, respectively. The increasing trend is illustrated in the box plot (Figure 5).

The effective diameter proved that the fictitious circumference of the NP was increased. However, several studies indicate that the encapsulated hydrophobic drug destabilises the isoform of the desolvated albumin, increasing the possibility of coalescence [8]. To assess whether the NPs lost their shape, we measured the shape factor (*Sf*). The average *Sf* for all the treatments was as follows: 0.85 ± 0.02, 0.87 ± 0.02, 0.89 ± 0.02, 0.84 ± 0.02, and 0.71 ± 0.07 at loading ratios of 0, 65, 130, 195, and 260 µg of capsaicin, respectively. The decreasing trend is illustrated in the box plot (Figure 6), indicating that the nanoparticles exhibited an increase in size, accompanied by a loss of their circular shape, as the concentration of capsaicin increased.

To complement the morphometric description of the NPs, the aspect ratio (*Ar*) was measured. *Ar* is a descriptor of the symmetry of the NP, i.e., it represents the proportionality between the major and minor axes of the particle and differs from the shape factor regarding describing the circularity of the particles. The average values of *Ar* were 1.06 ± 0.03, 1.06 ± 0.03, 1.05 ± 0.02, 1.07 ± 0.05, and 1.10 ± 0.08 at loading ratios of 0, 65, 130, 195, and 260 µg of capsaicin, respectively. The treatment showed that the best circularity of isolated nanoparticles was achieved using 0 and 65 µg of capsaicin (empty and low concentrations) (Figure 7). This morphology provides no assurance that the best internalisation occurs in the cells and tissue; however, entrapping the drug in the core of the NP could indicate the complete capacity to entrap.

### 3.4. ζ-Potential, Hydrodynamic Diameter, and PDI

The ζ-potential and hydrodynamic diameter are important parameters for measuring surface electrochemical charge, which influences stability and dispersion of NPs in suspension [39]. The hydrodynamic diameter refers to the apparent size of a particle in solution, considering the Brownian movement and interactions with the solvent [40]. Both parameters offer information about biodistribution and retention in the target tissues, and thus, affect nanoparticle toxicity [39,40].

The ζ-potential showed a decrease in electronegativity from empty NPs (0 µg of capsaicin loading ratio) to high capsaicin concentrations (260 µg of capsaicin loading ratio), with a strong correlation (*r* = 0.97, *p* = 0.0001) between the concentration of capsaicin added and the decrease in electronegative charge on the surface of the NP (Figure 8). This effect represents the opposite behaviour with respect to previous reports in which glutaraldehyde was used as the crosslinking agent and the concentration of capsaicin was 4 times less than that used in this study [8,9,10]. However, Choi et al. [5] and Ahmady et al. [41] found similar lower electronegativity values in capsaicin-loaded alginate NPs with a single layer synthesised via nanoemulsification at temperatures ranging from 85 to 100 °C. In both cases, the concentration of the capsaicin reached 10 mg/mL in the formulation, and the ζ-potential was −14.2 ± 1.96 and −24.9 ± 0.7 mV, respectively. The lower electronegativity is probably associated with greater quantities of encapsulated capsaicin, and BSA showed a dual function as an encapsulation matrix and stabilising agent, similar to the biopolymer Poloxamer 188 [40]. Additionally, according to several authors, the molecular interactions between albumin and hydrophobic compounds are mediated by hydrogen bonding and hydrophobic interactions, causing molecular reorganisation of the amino acids. This process exposes the hydrophobic sites in the albumin and increases the capacity to entrap hydrophobic drugs [8,42]. In this study, the use of heat (70 °C) as a crosslinking agent likely affected the albumin protein more than the capsaicin, since thermal oxidation of the capsaicin occurred at temperatures ranging from 265 to 425 °C [38]. Additionally, the capsaicin concentration used in this study was higher, facilitating the encapsulation of larger quantities of hydrophobic drugs.

The hydrodynamic diameters were 130.1 ± 0.60, 149.9 ± 2.59, 153.4 ± 2.40, 169.1 ± 1.33, and 249 ± 2.06 nm at loading ratios of 0, 65, 130, 195, and 260 µg of capsaicin, respectively (Figure 9). The hydrodynamic diameter is slightly larger than the effective diameter (*Ed*); this difference is due to the TEM image analysis, which reflects the dry state and internal core of the NPs. Dynamic light scattering (DLS) allows the measurement of NPs in solution, which allows the registration of the dynamic state of the corona, i.e., the functionalised surface of the NPs. The hydrodynamic diameter enables the determination of the correlation between particle size and physiological process [43]. This overestimation is probably due to the formation of a pseudo-corona composed of some fractions of the long hydrophobic chain, with a polar amide group being deposited on the surface of the nanoparticle, thereby increasing the size of the nanoparticles. A similar effect was reported on nanostructured lipid carriers [40]. Additionally, the super-translocation of capsaicin to the surface is the cause of the coalescence between particles and an increase in size at loading ratios of 195 and 260 µg of capsaicin. The mean PDI values were 0.06 ± 0.03, 0.08 ± 0.02, 0.05 ± 0.03, 0.06 ± 0.00, and 0.14 ± 0.05 at loading ratios of 0, 65, 130, 195, and 260 µg of capsaicin, respectively. The PDI was maintained below 0.4, indicating that the sample is not polydisperse; conversely, the distribution of NP size was homogeneous between treatments. The treatment with the highest formation of aggregates was that conducted using 260 µg of capsaicin.

### 3.5. Effect of Heat Crosslinking on Molecular Interactions Between Nanoparticles

The spectra of pure capsaicin and native BSA were analysed to enable the establishment of the original conditions of these molecules, as well as changes associated with the supramolecular assembling of albumin–capsaicin molecules and the effect of crosslinking, allowing further characterisation of NPs. The FTIR spectrum of capsaicin showed three specific bands that correspond to the raw molecule. The phenolic group (4-OH) was identified at 3443 cm^−1^. The region 3400–3200 cm^−1^ indicates a symmetric (sym) and asymmetric (asym) stretching of the hydroxyl group (O–H) [44]. Subsequently, the stretching amide bond (–NH) was identified at 3283 cm^−1^ and the aliphatic bond (C–H) was identified at 2922 cm^−1^. These peaks are important for evaluating molecular stability during the thermal crosslinking process. The alkyl C–N bond, which is connected to the amide group (–NH), is considered the most susceptible to molecular breakage during the thermal oxidation of capsaicin [38]. Subsequent peaks were observed at 1641 and 1518 cm^−1^, corresponding to alkane bonds (C–C) and carbonyl bonds (C–O), respectively. This study identified two scaled vibrational frequencies at 1415 and 1383 cm^−1^ that were assigned to CH_2_ scissoring and CH_2_ wagging vibrations, as described by Ou et al. [38] (Figure 10a, solid line).

The BSA native protein was analysed in previous works [9,10]. There are four characteristic peaks in BSA that correspond to amide functional groups and one peak corresponding to the carboxylic functional group. The largest peak at 3285 cm^−1^ corresponds to amide A and is associated with N–H stretching vibration [9]. While the C=O stretching corresponds to the amide I band at 1644 cm^−1^, a similar peak at 1515 cm^−1^ corresponds to the amide II band, and the molecular vibration is attributed to C–N stretching and N–H bending vibrations. The peak corresponding to the carboxylic functional group (CH_2_) occurred at 1393 cm^−1^. Ultimately, the identified amide peak observed at 1244 cm^−1^ corresponds to C–N stretching and N–H bending and was assigned to amide III. A similar spectrum was reported by Carriles et al. [10] (Figure 10a, dotted line).

The spectrum of BSA NPs (0 µg of capsaicin) and BSA–capsaicin (loading ratios of 16.2, 32.5, 48.7, and 65.0 µg of capsaicin) showed several differences in all peaks previously reported by Sanchez-Arreguin et al. [9] and Carriles et al. [10]. The first deformed peak (broadening peak) occurred between 3648 and 3111 cm^−1^, with a maximal intensity at 3277 cm^−1^, and corresponded to the N–H stretching vibration of amide A in BSA (blue rectangle in Figure 10b) and overlapped the 4-OH group and NH amide bond in capsaicin (green rectangle in Figure 10b) from 3449 to 3154 cm^−1^. It is important to empathise that the NH stretching frequencies in the 4-OH group occur in this same spectral range, hence the predominance of NH stretching interference during the identification of the OH group [45]. However, for all capsaicin treatments, the NH amide bond signal in pure capsaicin (N–H stretching) in the region from 3404 to 3216 cm^−1^ showed a progressive decrease (Figure 10b). This result was contrary to that obtained with glutaraldehyde-crosslinked NPs.

A similar effect was observed in the region between 1722 and 1452 cm^−1^, corresponding to an overlap between amide I and II bands of BSA and the alkane bond (C–C) and carbonyl bond (C=O) of the hydrophobic side chain of capsaicin (1710–1480 cm^−1^). The region from 1415 to 1366 cm^−1^ corresponds to the CH_2_ scissoring and CH_2_ wagging peak of capsaicin, and both overlap with the nanostructuration of BSA. Razzak and Cho [42] found a similar effect when ovalbumin and lactalbumin produced a molecular interaction with capsaicin. They attributed this to a specific interaction between the amino functional groups and the hydrophobic chains in capsaicin. Therefore, the effect of the increase in capsaicin in the formulation could be determined, although there were changes in the intensity of the overlapping peaks.

### 3.6. Raman Analysis

Raman analysis is a powerful tool for determining the changes in the molecular structure of functional groups. The similarity between FTIR and Raman is often overstated due to the similarity of molecular vibrational frequencies and working scales (spectra), although the vibrational band intensities differ [45]. However, it is inaccurate to consider the methods as analogues; each analysis method provides different results and is complementary rather than competitive. To characterise the native molecules, capsaicin and BSA were analysed. Capsaicin had an absorption band above 3000 cm^−1^ that corresponds to O–H–N–H and C–H stretching. The band corresponding to the O–H bond was observed at 3295 cm^−1^ (Figure 11a); this hydrogen bond is important as it reveals details and changes in the structure [45,46]. Capsaicin showed a heterocyclic aromatic compound in the form of vanillylamine, synthesised from vanillin [47]. The –CH stretching vibration bands are characteristic of the heteroaromatic structures and occur between 3000 and 3100 cm^−1^ [46,48]. However, in this study, CH bands were observed at 2871, 2887, 2904, 2935, 2954, 3024, and 3068 cm^−1^; the proximity of the vibration is probably due to the benzene ring (aromatic ring) (Figure 11a). The bands comprise two peaks from 1657 to 1598 cm^−1^ correspond to C=O double bond stretching; subsequently, the absorbed bands occurring from 1400 to 1600 cm^−1^ correspond to the C–C stretching vibration (Figure 11c). These bands are characterised by two similar peaks at 1462 and 1439 cm^−1^ corresponding to symmetrical vibrations. The –CH out-of-plane was observed at 807 cm^−1^ modes of the chain (Figure 11c).

On the other hand, BSA exhibited a fingerprint from 2000 to 400 cm^−1^ in the Raman spectrum. The molecular interactions are due to the secondary and tertiary structures and the microenvironment of each functional group [49]. Chikashi and Kazufumi [49] reported that the –CH and –OH stretching bands are sensitive to changes in the geometry of hydrogen bonds and cause peak deformations from 2800 to 4000 cm^−1^. In native BSA (lyophilised), the peak at 2931 cm^−1^ was assigned to the –CH stretching band (Figure 11b). The peak at 1659 cm^−1^ corresponds to the amide I band, and the –CH_2_ stretching band was observed at 1450 cm^−1^. The phenyl ring angular bending vibrations corresponding to the hydrophobic region of phenylalanine (Phe) were observed at 1007 cm^−1^, and the side by side doublet Y_5_–Y_6_ at 855 cm^−1^ probably correspond to the Cα–Cβ of amino bond to the aromatic side chain orientation (phenyl ring) bond of the main chain backbone tyrosine (Tyr) (Figure 11d) [50,51,52].

The change in the concentration of one compound or the reorganisation of several compounds under the matrix was determined using Raman spectroscopy. In the case of BSA–capsaicin NPs, two regions, from 3400 to 2700 cm^−1^ and 1700 to 50 cm^−1^, showed spectral bands associated with the reorganisation of the BSA molecules to increase the capacity to entrap capsaicin. The first region showed a gradual reduction in the intensity of the Raman signal associated with the –CH stretching band at 2934 cm^−1^ (Figure 11e). The peak of the empty nanoparticles (0 µg of capsaicin) was similar to that of native BSA (Figure 11e, black line); however, the peak flattened as the concentration of capsaicin increased; this behaviour is associated with the reorganisation of capsaicin on the surface of the nanoparticles due to the concentration of BSA remaining constant. This effect was reported by Chikashi and Kazufumi [49]. They found that the Raman spectra of the –CH stretching regions (2930 cm^−1^) of BSA increased in intensity with the concentration of BSA (20, 40, 100, 200, and 300 mg/mL). The second region showed similar changes to those previously mentioned. The spectrum of the empty nanoparticles showed a decrease in the intensity of the Raman signal as the capsaicin concentration increased. The functional groups that were affected are the amide I band (1659 cm^−1^), the –CH_2_ stretching band (1450 cm^−1^), and hydrophobic residues of phenylalanine and tyrosine at 1007 and 855 cm^−1^, respectively (Figure 11f).

In this study, the molecular interactions between BSA and capsaicin are predominantly facilitated by five amino acids: Tyr400–capsaicin, Asn401–capsaicin, Lys524–capsaicin, Phe506–capsaicin, and Phe550–capsaicin [10,53]. Phenylalanine and tyrosine may bind strongly with capsaicin via hydrophobic interaction. In contrast, glutaraldehyde crosslinking involves specific formation of covalent bonds with unprotonated amino groups, such as the ε-amino groups of lysine, which are very reactive nucleophilic agents [20,54].

### 3.7. SDS-PAGE of the NPs

Although SDS-PAGE is commonly used to determine the molecular size and purity of proteins, this technique has recently been applied in the analysis and comparison of the effect of crosslinking agents in the formulation of BSA NPs [55]. An efficient crosslinking mechanism protects albumin NPs from disassembly in a wide range of environments. BSA yielded a broad band in the 66–72 kDa region (Figure 12, lane 2). To demonstrate heat-based crosslinking, a similar quantity of BSA from NP formulations was mixed with capsaicin and dissolved in water at pH 10 without heat-based crosslinking. The band had a similar molecular weight (MW) and thickness to native BSA (Figure 12, lane 3), which indicates that the albumin does not allow crosslinking. A second control—with a similar formulation to the previous treatment, but with 16 mL of ethanol—was placed in the gel without heat-based crosslinking, and the band was not detected, probably because albumin was denatured (Figure 12, lane 4). The samples were placed in the gel in ascending order of capsaicin concentration from 0 to 260 µg (Figure 12, lanes 5 to 9). The band pattern showed that NPs allowed the disassembly into the original units of bovine serum albumin at 66 kDa. Additionally, the breadth of the band increased with the concentration of capsaicin; however, albumin concentration remained constant across all treatments. This effect was probably due to the high concentration of capsaicin increasing the propensity of BSA NPs to disassemble under SDS-PAGE conditions. These results contrast with those of Carriles et al. [10], who reported that BSA–capsaicin NPs crosslinked with glutaraldehyde did not disassemble, even when treated with SDS, heat, and β-mercaptoethanol.

### 3.8. Biocompatibility of BSA–Capsaicin In Vivo

We conducted a pilot study to assess whether the BSA–capsaicin nanoparticles caused an inflammatory reaction in the liver. To this end, liver tissue sections exposed to treatments with 0, 65, and 260 µg of capsaicin (low and high capsaicin loading ratios) and a control (10 mM NaCl) were subjected to histopathological analysis. In the control treatment (Figure 13a), the cell morphology exhibited a normal appearance, with hepatocytes arranged in an orderly manner. Some cells showed empty areas, corresponding to normal storage of lipids (black arrow). The hepatocytes did not show symptoms of inflammation and necrosis. In contrast, the group of mice injected with empty BSA nanoparticles showed alterations in liver tissue. The accumulation of fat (steatosis) was similar to that of the control (Figure 13b) and exhibited infiltration of erythrocytes through to hepatic sinusoids, with some vessels showing an accumulation of blood (black arrows). The treatment with a low capsaicin loading ratio (65 µg) had a beneficial effect on the hepatocytes, as no erythrocyte infiltration was observed in the hepatic sinusoids, and the size of these structures was smaller than that of the control and empty nanoparticles. Additionally, erythrocytes did not accumulate in the blood vessels (Figure 13c). The most significant impact was probably the reduction in the level of lipids in hepatocytes (decreased steatosis). Treatment with 260 µg of capsaicin (high capsaicin loading ratio) also resulted in a decrease in steatosis (Figure 13d); however, a change in the size of hepatic sinusoids was observed (black arrow), and within these spaces, immune cells, such as macrophages (white arrow), were present. This is an indication of an inflammatory process.

To verify the reduction in lipid droplets, the liver tissues were analysed using multiphoton microscopy. Intracellular lipid droplets were stained with Nile red and counterstained with anti-Actin488 nm and DAPI to visualise the nuclei. The control treatment showed normal accumulation of lipids resulting from the ingestion of food (Figure 14a, white arrow, droplets in red). As mentioned in Section 2, the mice were not subject to food restriction; hence, they may gain weight due to increased food consumption. The treatment consisting of empty nanoparticles (0 µg of capsaicin loading ratio) showed a similar accumulation of lipid droplets; however, some symptoms, such as the increase in the actin signal and the presence of hepatocytes with binucleated nuclei (white arrows), are indicators of the toxicity of albumin nanoparticles. Treatment with low doses of capsaicin (65 µg of capsaicin loading ratio) resulted in five fat droplets (white arrow, structures stained red) in all hepatocytes within the region of interest (ROI) (Figure 14c). Additionally, no nuclear alterations were observed, and the cytoskeletal actin remained intact. Finally, the treatment with a higher concentration of capsaicin (260 µg of capsaicin loading ratio) showed three fat droplets (white arrows, structures stained red) within the region of interest (ROI) in all tissues analysed (Figure 14d). However, the presence of the binuclear hepatocytes (orange arrows) and the increase in the size of the hepatic sinusoids indicated an inflammatory response.

The effect of empty BSA nanoparticles observed in this study showed some similarities to those reported by Da Silva et al. [56], who analysed histopathological skin sections of mice treated with BSA NPs. They found increased epidermal thickness, haemorrhage, and the presence of inflammatory cells (mononuclear cells) seven days after administration. However, the pro-inflammatory effect of BSA nanoparticles depends on the dose and type of administration. The doses used in this study may not have reached the threshold required to trigger several inflammatory effects. On the other hand, the decrease in the number of lipid droplets in treatments containing 65 and 260 µg of capsaicin loading ratio may be due to the thermogenic effect of the capsaicin. Capsaicin induces ATP-dependent thermogenesis through receptors TRPV1, β3-AR, and α1-AR in 3T3-L1 in mice [57]. Several studies have reported that dietary capsaicin attenuates obesity, hepatic steatosis, adipocyte size, and inflammation [58,59,60]. However, this study is the first to report a pharmacological effect of nanoencapsulated capsaicin administered through a different route, such as ingestion through food. As a caveat, we acknowledge that the sample size of the mice in our study is limited. Although the results are clear, they should be considered preliminary data on which to build larger and more detailed future studies.

## 4. Conclusions

Our work demonstrated that the use of heat as a crosslinking agent in the synthesis of BSA–capsaicin NPs improves the morphology, morphometric parameters, and ζ-potential. FTIR and RAMAN spectroscopy confirmed that the molecular interactions between BSA and capsaicin were established from 3400 to 2700 cm^−1^ and 1700 to 50 cm^−1^. Both spectral bands were associated with the reorganisation of BSA molecules to increase the capacity to entrap capsaicin. Additionally, the BSA–capsaicin nanoparticles showed a great capacity to entrap hydrophobic drugs, reaching above 22.9 ± 1.32 mg at 31.2 mg of initial capsaicin. The BSA nanostructured and stabilised with heat crosslinking showed a capacity to denature in BSA at 66 kD. Treatment without heat-based crosslinking of BSA was not detected by SDS-PAGE; on the other hand, the bioactivity of BSA–capsaicin nanoparticles at low and higher capsaicin loading ratios had a beneficial effect in hepatocytes, decreasing the accumulation of lipid droplets. However, formulations with high concentrations of capsaicin increased BSA instability and resulted in protein denaturation, decreasing the yield of NPs. These findings support the idea that heat can be used as a crosslinking alternative to glutaraldehyde in the formulation of BSA nanoparticles for encapsulating hydrophobic drugs such as capsaicin. This method of crosslinking improved the properties of nanoparticles and enhanced parameters important in the pharmaceutical industry, such as safety.

## Figures and Tables

**Figure 1 pharmaceutics-17-01306-f001:**
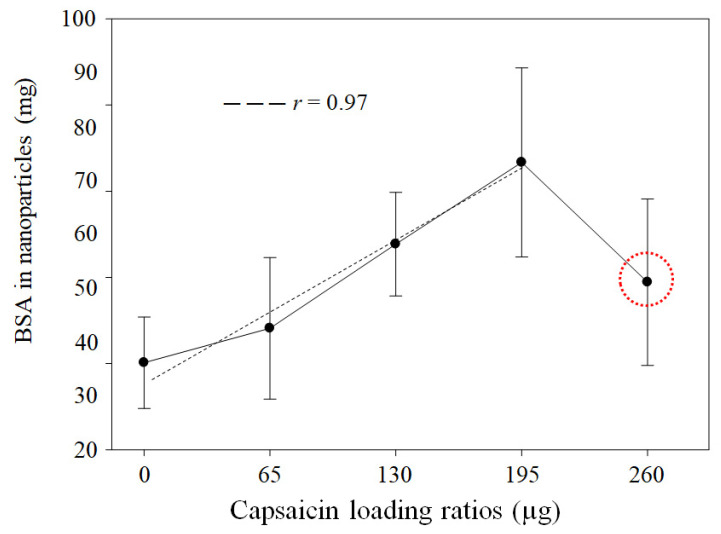
Effects of heat crosslinking and capsaicin dose on BSA NP formulation. The quantified protein shows a linear trend (*r* = 0.97, *p* = 0.0003) based on the concentration of capsaicin (from 0 to 195 µg) added to the formulation. Higher capsaicin concentration (loading ratio of 260 µg of capsaicin) (red circle) shows an adverse effect of heat crosslinking. Values are means of three experimental replicates ± standard deviations.

**Figure 2 pharmaceutics-17-01306-f002:**
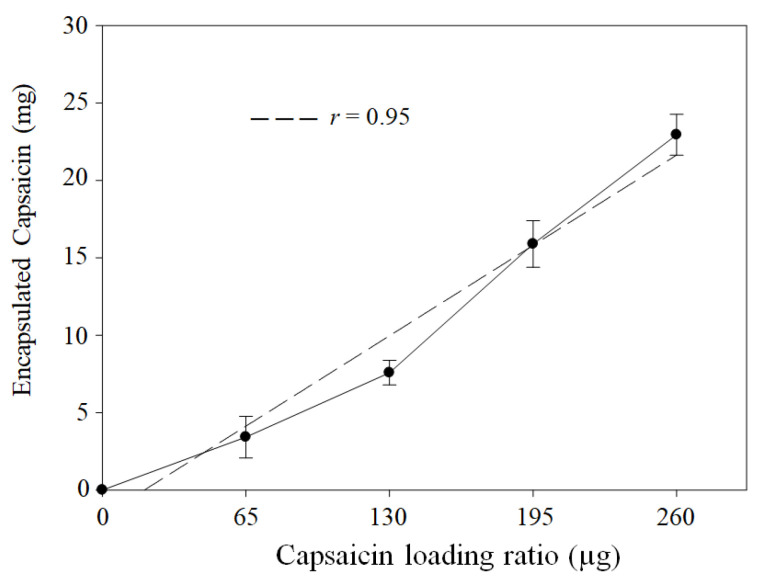
Heat-crosslinked encapsulation and capsaicin in NPs. The quantified capsaicin concentration showed a linear trend (*r* = 0.95, *p* = 0.0003). Values are means of three experimental replicates ± standard deviations.

**Figure 3 pharmaceutics-17-01306-f003:**
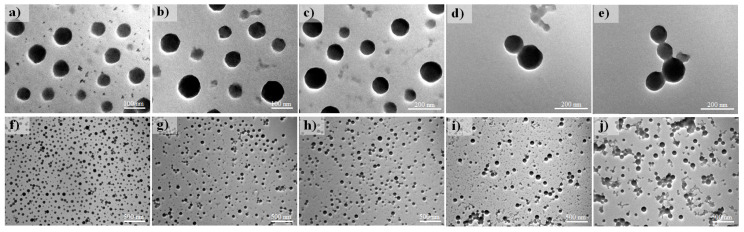
NP analysis using TEM. TEM micrographs obtained at high magnification show isolated NPs with detailed morphology and nanostructures (**a**–**e**). TEM micrographs obtained at low magnification show the size and shape distribution of BSA–capsaicin nanoparticles at different capsaicin concentrations (**f**–**j**). All TEM experiments were conducted at 80 kV high voltage (EHT), 710,000× to 140,000× (high magnification), 8900× to 9000× (low magnification), and 5 × 10^−3^ Pa (5 × 10^−5^ Torr) working pressure.

**Figure 4 pharmaceutics-17-01306-f004:**
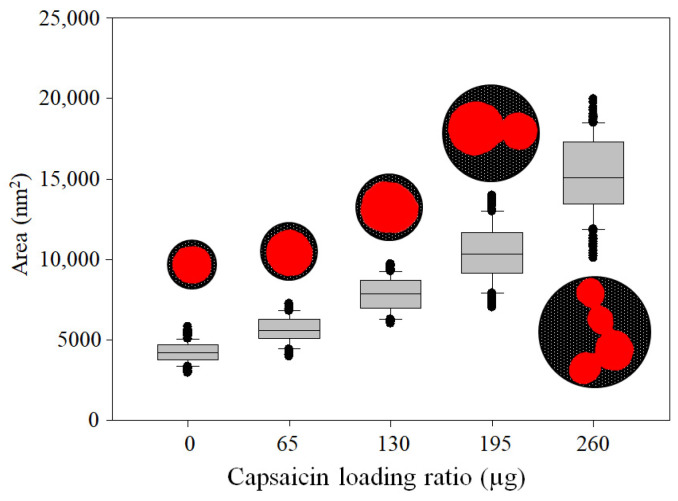
Effects of heat crosslinking and capsaicin concentration on NP area. The NP area showed a tendency to increase with the concentration of capsaicin. Conditions: digital measurement of n = 180 nanoparticle images at 1376 × 1032 pixels and 8-bit compression; resolution set at 2.1 pixels/nm. Values are the means of three experimental replicates ± standard deviations.

**Figure 5 pharmaceutics-17-01306-f005:**
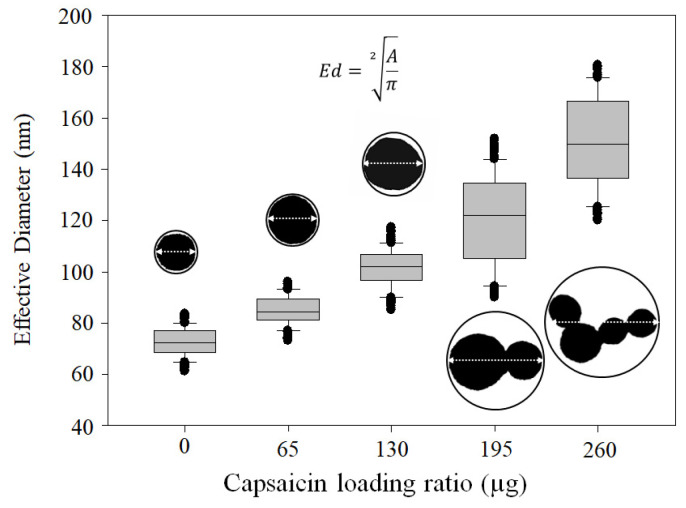
Effects of heat crosslinking and capsaicin on NP diameter. The effective diameter of the isolated NPs increased with increasing capsaicin concentration. Conditions: digital measurement of n = 180 nanoparticle images at 1376 × 1032 pixels and 8-bit compression; resolution set at 2.1 pixels/nm. Values are the means of three experimental replicates ± standard deviations.

**Figure 6 pharmaceutics-17-01306-f006:**
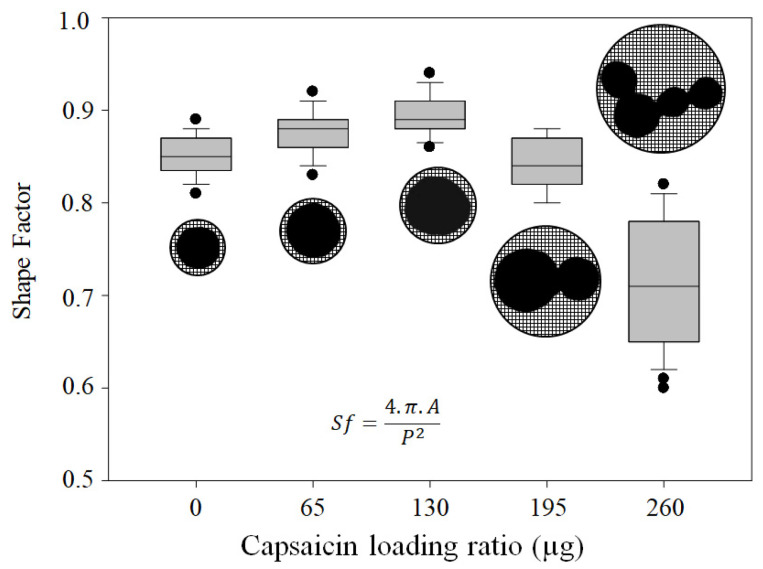
Effects of heat crosslinking and capsaicin on NP shape factor. The shape factor of the isolated NPs demonstrated the best circular shape at a loading ratio of 130 μg of capsaicin. Increasing the concentration of capsaicin led to a loss of shape. Conditions: digital measurement of n = 180 nanoparticle images at 1376 × 1032 pixels and 8-bit compression; resolution set at 2.1 pixels/nm. Values are the means of three experimental replicates ± standard deviations.

**Figure 7 pharmaceutics-17-01306-f007:**
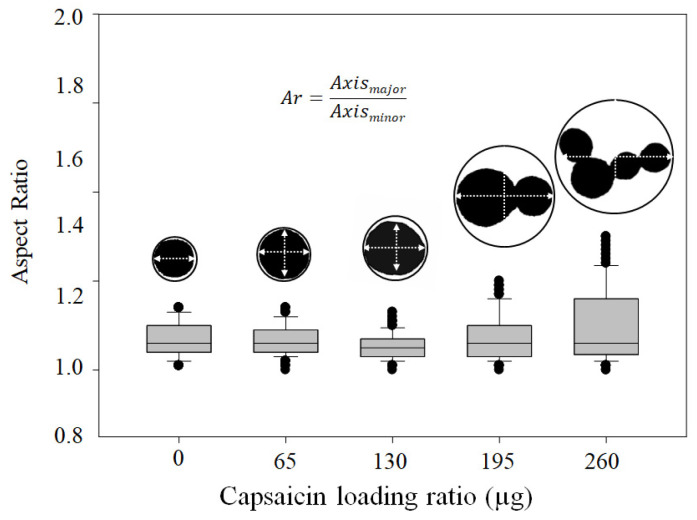
Effects of heat crosslinking and capsaicin on NP aspect ratio. The aspect ratio of NPs showed a gradual transition from irregular dimension (*X*-*Y* axis) to circular shape and irregular aspect at high capsaicin concentration. Conditions: digital measurement of n = 180 nanoparticle images at 1376 × 1032 pixels and 8-bit compression; resolution set at 2.1 pixels/nm. Values are the means of three experimental replicates ± standard deviations.

**Figure 8 pharmaceutics-17-01306-f008:**
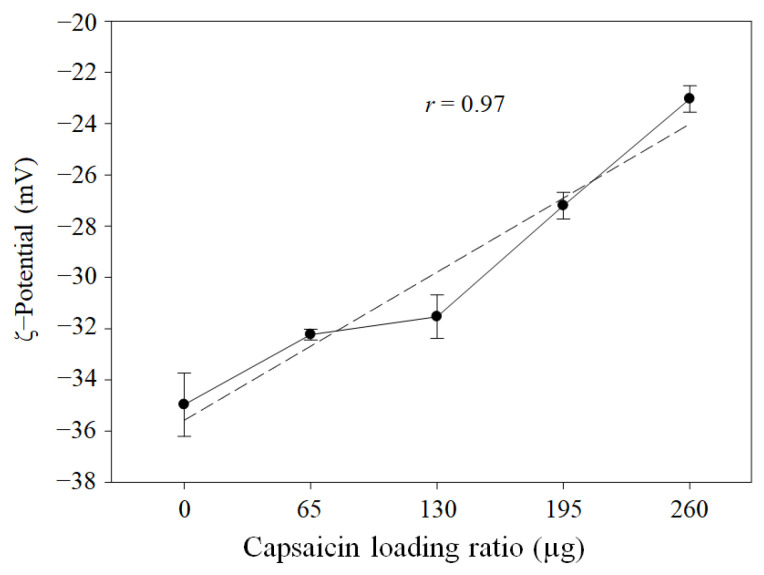
Effects of heat crosslinking and capsaicin on NP ζ-potential. The ζ-potential of BSA–capsaicin NP showed a decrease in electronegativity with increasing capsaicin concentration. Values are means of three experimental replicates  ±  standard deviations.

**Figure 9 pharmaceutics-17-01306-f009:**
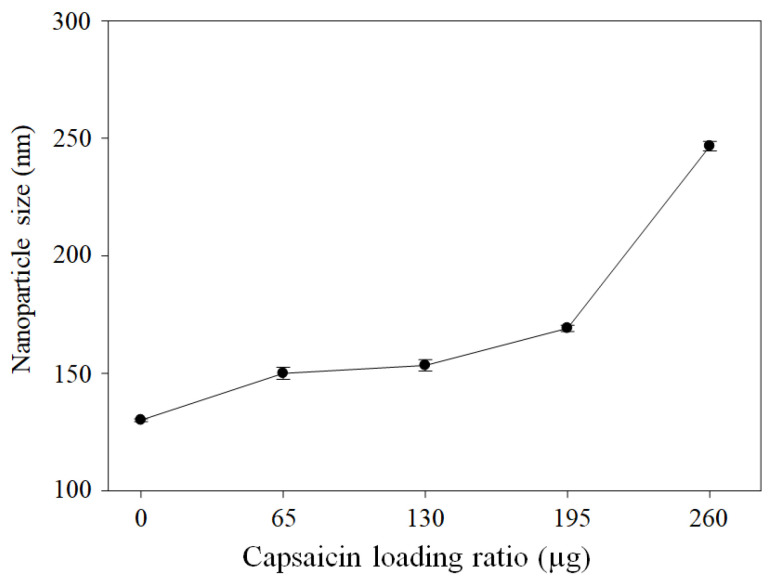
Effects of heat crosslinking and capsaicin on NP hydrodynamic diameter. The hydrodynamic diameters of BSA–capsaicin NPs showed an association between increased size and capsaicin concentration. Values are means of three experimental replicates  ±  standard deviations.

**Figure 10 pharmaceutics-17-01306-f010:**
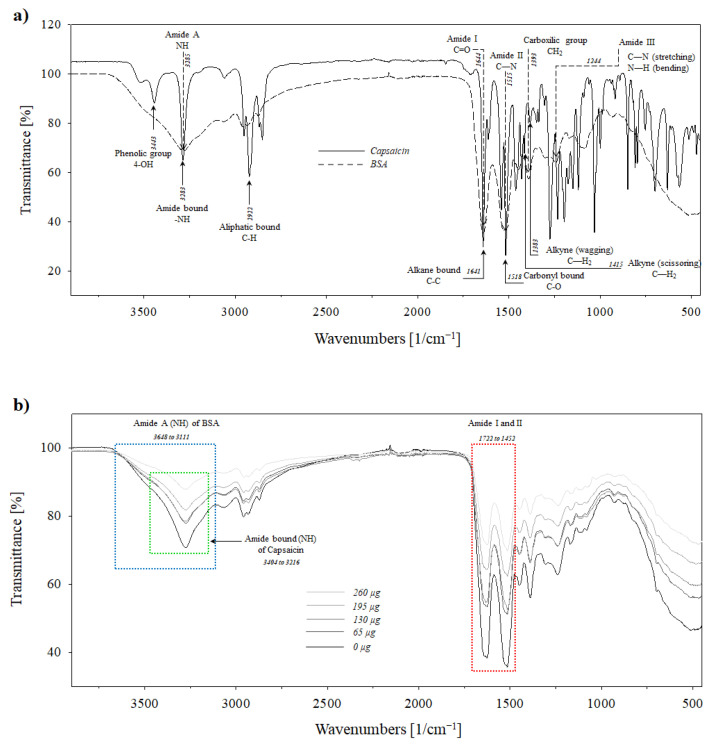
Molecular interactions between BSA and capsaicin molecules and the effect of heat crosslinking on NP synthesis. (**a**) FTIR spectra of capsaicin (solid line) and BSA before heat crosslinking (dotted line). (**b**) FTIR spectra of BSA–capsaicin NPs at 0, 65, 130, 195, and 260 µg of capsaicin loading ratios (lines in gradient of grey) after heat crosslinking. The graphs show deformation of the peak that corresponds to N–H stretching vibration of amide A of BSA (blue rectangle) between 3648 and 3111 cm^−1^. The second region, from 1722 to 1452 cm^−1^ (red rectangle), corresponds to the interaction between amide I and amide II of BSA, with the hydrophobic side chain of capsaicin. The scanning spectral range was 400–4000 cm^−1^ in transmittance mode.

**Figure 11 pharmaceutics-17-01306-f011:**
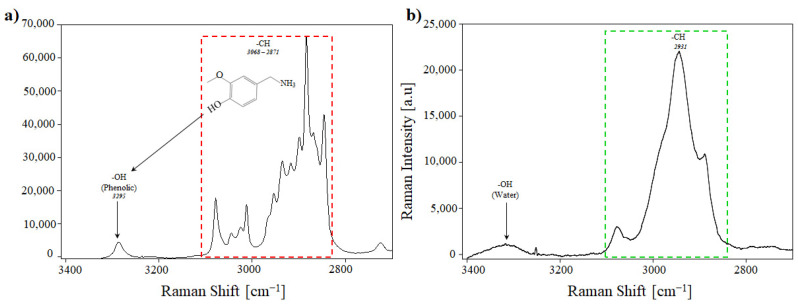
RAMAN analysis of the interaction between BSA and capsaicin and the effect of heat crosslinking on NP synthesis. (**a**) RAMAN spectrum of capsaicin, showing O–H and –CH stretching vibrations (red rectangle). (**b**) In the same region, BSA shows a –CH stretching band (green rectangle). (**c**) Fingerprint region of capsaicin showing C=O double bond stretching, C–C stretching vibration, and –CH out-of-plane (red rectangle). (**d**) Fingerprint region of BSA showing the amide I band, the –CH_2_, and the hydrophobic region of phenylalanine (Phe) and tyrosine (Tyr) (green rectangle). RAMAN spectra of BSA–capsaicin NPs at 0, 65, 130, 195, and 260 µg of capsaicin loading ratios (lines in gradient of grey) after heat crosslinking. (**e**) First region, from 3400 to 2700 cm^−1^, is associated with the –CH stretching band at 2934 cm^−1^. (**f**) Second region, from 1700 to 50 cm^−1^, corresponds to the amide I band (1659 cm^−1^), the –CH_2_ stretching band (1450 cm^−1^), and hydrophobic residues of phenylalanine and tyrosine at 1007 cm^−1^ and 855 cm^−1^.

**Figure 12 pharmaceutics-17-01306-f012:**
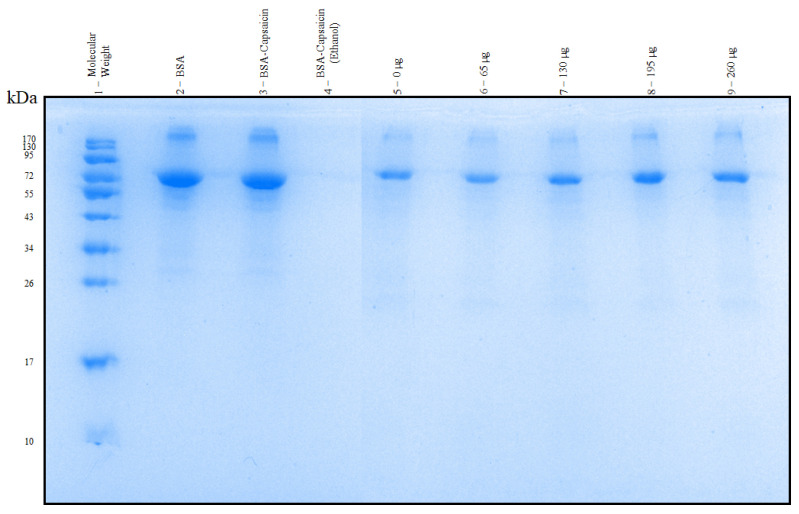
Electrophoretic patterns of disassembled BSA–capsaicin NPs. Lane 1 shows the molecular weight (MW) ladder, from 10 to 170 kDa. Line 2 is the positive control (native BSA). Line 3 is the BSA–capsaicin without heat crosslinking. Line 4 is the BSA–capsaicin and ethanol without heat crosslinking. Lines 5–9 represent NPs formulated at 0, 65, 130, 195, and 260 µg of capsaicin loading ratios.

**Figure 13 pharmaceutics-17-01306-f013:**
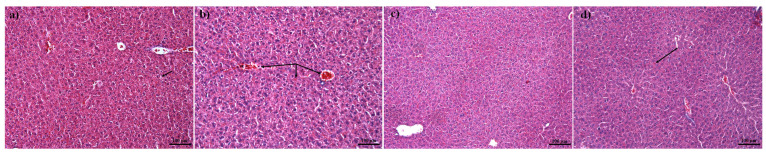
Histopathological analysis of liver tissue. (**a**) Control treatment showing the normal appearance of hepatocytes with normal lipid storage (black arrow). (**b**) Hepatocytes showing steatosis and erythrocyte infiltration into hepatic sinusoids (black arrows). (**c**) At a low capsaicin loading ratio (65 µg), the hepatocytes show reduced steatosis, and blood vessels show no haemorrhage. (**d**) Treatment with a loading ratio of 260 µg of capsaicin has an antisteatotic effect. Inflammatory symptoms are observed, including increased hepatic sinusoids (black arrow) and the presence of immune cells (white arrow).

**Figure 14 pharmaceutics-17-01306-f014:**
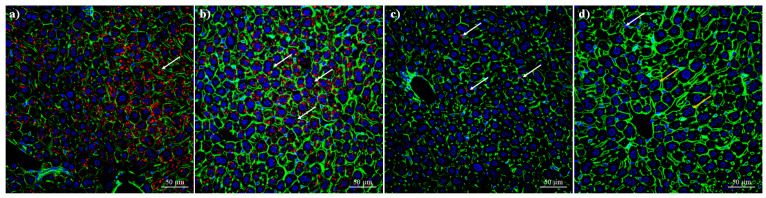
Analysis of lipid droplets using multiphoton microscopy. (**a**) Control treatment showing normal accumulation of lipid droplets. (**b**) Treatment with 0 µg of capsaicin shows similar accumulation of lipid droplets as the control; however, toxicity of the empty nanoparticles is observed, including increased actin and binucleated hepatocytes (white arrows). (**c**) Treatment with 65 µg of capsaicin shows decreased accumulation of lipid droplets (white arrow, structures stained red); additionally, no nuclear alterations are observed, and the cytoskeleton-derived actin remains intact. (**d**) Treatment with 260 µg of capsaicin shows a similar effect regarding fat droplets (white arrows, structures stained red); however, symptoms associated with inflammation are observed, including the presence of binuclear hepatocytes (orange arrows) and an increase in the size of the hepatic sinusoids.

## Data Availability

Data will be made available upon request.

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
