# Peer review of "Effect of Crosslinking Using Heat on the Physicochemical Features of Bsa–Capsaicin Nanoparticles"

_pharmaceutics, 2025, doi:10.3390/pharmaceutics17101306_

Round 1
Reviewer 1 Report (Previous Reviewer 2)
Comments and Suggestions for Authors
The reviewer appreciates the author's effort in revising the manuscript. I have no further comments.
Author Response
We appreciate the review and comments of Reviewer 1.

Reviewer 2 Report (Previous Reviewer 1)
Comments and Suggestions for Authors
This study addresses an important question in pharmaceutical nanotechnology: whether heat treatment can effectively replace glutaraldehyde in the synthesis of BSA-capsaicin nanoparticles. The topic is timely and relevant in terms of reducing toxic crosslinkers and improving formulation safety.
This study combined nanoparticle synthesis, physicochemical characterization, and preliminary in vivo evaluation, providing a potentially valuable contribution. However, I identified some points where the interpretation needs to be refined, the calculations checked, and the conclusions worded more cautiously. These revisions do not invalidate the work but are essential for improving accuracy and scientific rigor.
Major Concerns
Question 1. Statistical correlation (p.7): The manuscript reports correlation coefficients of r = 0.97 and r = 0.95, both with p = 0.05, based on five data points (capsaicin concentrations 65–390 µg). For such high r-values and n = 5, the corresponding p-value is typically much smaller (≈0.003). This suggests a possible miscalculation or a transcription error. I recommend revising the reported values to ensure internal statistical consistency in the manuscript.
Question 2. BSA isoform interpretation (p.7): The reduced protein yield at higher capsaicin concentrations is attributed to a “reversibility of BSA isoforms A to B/N.” Although interesting as a hypothesis, this explanation is not directly supported by the experimental data presented, nor is it referenced with robust evidence for the specific conditions used (pH 10 and 70 °C). Reframing this interpretation as a plausible hypothesis rather than a definitive explanation would strengthen the manuscript.
Question 3. Zeta potential and colloidal stability (p.12): The reduction of ζ-potential from –34 mV to –24 mV is interpreted as “not compromising stability.” However, according to accepted colloid science, absolute values below ±30 mV indicate only moderate or limited stability. Moreover, attributing the reduction in ζ to an increase in the particle diameter is conceptually inaccurate because ζ reflects the surface charge density rather than the particle size. I suggest revising the discussion to better align with the established colloid literature.
Question 4. Polydispersity Index (PDI) interpretation (p.13): The manuscript states that a PDI above 0.4 “indicates that the sample is not polydisperse,” while in fact ISO 22412 and DLS literature classify PDI values >0.4 as indicative of highly polydisperse, heterogeneous samples. This represents a significant conceptual inversion. The text should be corrected to avoid misleading the readers and to comply with standard definitions.
Question 5. SDS‑PAGE interpretation (p.17): SDS‑PAGE results are described as evidence that the nanoparticles “disassembled” into 66 kDa BSA subunits. However, because SDS-PAGE inherently applies denaturing conditions (SDS, β‑mercaptoethanol, heat), BSA will always appear at 66 kDa, regardless of supramolecular organization. The results confirmed the presence of BSA but did not demonstrate the controlled disassembly of nanoparticles. The interpretation would benefit from rephrasing to avoid overstating what the assay proves.
Question 6. Sample size in in vivo studies (p.18–19): The in vivo evaluation used only 12 animals divided into multiple groups. While the findings are suggestive, such a small cohort limits the statistical power and generalizability. I recommend explicitly acknowledging this limitation in the discussion section so that readers can interpret the data with appropriate caution.
Question 7. Generalization in the conclusion (p.20): The conclusion extends the findings with capsaicin to “other hydrophobic drugs.” As no other compounds were tested, this suggestion should be presented with caution. Rephrasing the statement as a hypothesis for future validation, rather than as an established applicability, would improve the accuracy of the conclusions drawn.
Author Response
Question 1. Statistical correlation (p.7): The manuscript reports correlation coefficients of r = 0.97 and r = 0.95, both with p = 0.05, based on five data points (capsaicin concentrations 65–390 µg). For such high r-values and n = 5, the corresponding p-value is typically much smaller (≈0.003). This suggests a possible miscalculation or a transcription error. I recommend revising the reported values to ensure internal statistical consistency in the manuscript.
Response: We consider the observation of Reviewer 2 to be valuable, and we sincerely appreciate the contribution to improving the quality of the manuscript. The statistical reports indicated p = 0.003 values for both probes. We admit it was a transcription error. We reiterate our thanks and appreciation to Reviewer 1 for the valuable observation.
According to the comment of Reviewer 2, we changed the p-values in sections: Abstract, Results and Discussion, Subsection: 3.1 Yield of BSA Transformed to NP and Quantification of Loaded Capsaicin, and Figure Captions:
Original version (Lines from 33-36):
The initial concentration of BSA and capsaicin in the formulation were directly correlated with the amount of BSA that was transformed into nanoparticles and loaded capsaicin (r=0.97, p=0.05 and r=0.95, p=0.05), respectively.
The sentences were changed as follows (Lines from 33-36):
The initial concentrations of BSA and capsaicin in the formulation were directly correlated with the amount of BSA that was transformed into nanoparticles and the loaded capsaicin (r=0.97, p=0.0003 and r=0.95, p=0.0003), respectively.
Original version (Lines from 287-289):
The quantities of native BSA that were transformed into nanoparticles increased with initial capsaicin concentration (treatments from 0 to 195 µg of loading ratio of capsaicin), showing a linear tendency (r=0.97, p=0.05) (Figure 1).
The sentences were changed as follows (Lines from 290-289):
The quantities of native BSA transformed into nanoparticles increased with initial capsaicin concentration (treatments ranging from 0 to 195 µg of capsaicin loading ratio), showing a linear trend (r=0.97, p=0.0003) (Figure 1).
Original version (Lines from 318-319):
On the other hand, the quantification of the capsaicin loaded into the nanoparticles showed a linear correlation (r=0.95, p=0.05) with initial capsaicin concentration (Figure 2).
The sentences were changed as follows (Lines from 322-323):
On the other hand, quantification of the capsaicin loaded into the nanoparticles showed a linear correlation (r=0.95, p=0.0003) with initial capsaicin concentration (Figure 2).
Original version (Lines from 877-879):
Figure 1. Effect of heat-crosslink and capsaicin dose on BSA NP formation. The quantified protein showed a linear tendency (r=0.97, p=0. 0.05) from 0 to 195 µg of capsaicin loading ratio added to formulation.
The sentences were changed as follows (Lines from 880-882):
Figure 1. Effects of heat crosslinking and capsaicin dose on BSA NP formulation. The quantified protein shows a linear trend (r=0.97, p=0.0003) based on the concentration of capsaicin (from 0 to 195 µg) added to the formulation.
Original version (Lines from 882-884):
Figure 2. Encapsulation of heat-crosslinked and capsaicin in NP. The quantified capsaicin showed a linear tendency (r=0.95, p=0.005). Values are means of three experimental replicates ± standard deviations.
The sentences were changed as follows (Lines from 885-887):
Figure 2. Heat-crosslinked encapsulation and capsaicin in NPs. The quantified capsaicin concentration showed a linear trend (r=0.95, p=0.0003). Values are means of three experimental replicates ± standard deviations.
Question 2. BSA isoform interpretation (p.7): The reduced protein yield at higher capsaicin concentrations is attributed to a “reversibility of BSA isoforms A to B/N.” Although interesting as a hypothesis, this explanation is not directly supported by the experimental data presented, nor is it referenced with robust evidence for the specific conditions used (pH 10 and 70 °C). Reframing this interpretation as a plausible hypothesis rather than a definitive explanation would strengthen the manuscript.
Response: We consider the observation of Reviewer 2 to be valuable, and we sincerely appreciate the contribution to improving the quality of the manuscript. In the manuscript, we propose that the change in albumin isoforms is one of the factors responsible for the decrease in the yield of BSA transformed into nanoparticles. However, we do not have experimental data that supports this hypothesis. We have several (unpublished) assays that demonstrate the influence of pH (from pH 7 to 11) in nanoparticle formation using BSA with different purification methods, such as Cohn and heat shock methods. However, we consider that this information is not relevant to be published. Therefore, we only use this information to control the variables that could affect the assembly of albumin nanoparticles. However, to clarify this point, we restructure the redaction of this sentence.
Original version (Lines from 291-294):
The decrease in the native BSA transformed into nanoparticles was probably caused by the capacity of reversibility from isoform A to isoform B or N of BSA by a change of basic pH [30]. The change of pH was probably caused by a higher concentration of capsaicin.
The sentence was changed as follows (Manuscript corrected, Lines 293-298):
The native BSA showed the capacity to change from isoform A to isoform B or N following a change in the basic pH [30]. Based on these facts, we propose the hypothesis that the decrease in native BSA transformed into nanoparticles was probably caused by the reversibility of BSA isoform A to N. The change in pH was probably caused by a higher concentration of capsaicin.
Question 3. Zeta potential and colloidal stability (p.12): The reduction of ζ-potential from –34 mV to –24 mV is interpreted as “not compromising stability.” However, according to accepted colloid science, absolute values below ±30 mV indicate only moderate or limited stability. Moreover, attributing the reduction in ζ to an increase in the particle diameter is conceptually inaccurate because ζ reflects the surface charge density rather than the particle size. I suggest revising the discussion to better align with the established colloid literature.
Response: We consider the observation of Reviewer 2 to be valuable, and we sincerely appreciate the contribution to improving the quality of the manuscript “In the section 3.4 ζ-potential, Hydrodynamic Diameter and PDI”, in the third paragraph, the sentence with the with the follow statement: “On the other hand, the hydrodynamic diameter showed a similar trend to ζ-potential, likely due to the interdependence of the two parameters.” This sentence confuses; hence, we remove this assertion.
Original version (Lines from 429-430):
On the other hand, the hydrodynamic diameter showed a similar trend to ζ-potential, likely due to the interdependence of the two parameters. The hydrodynamic diameter was 130.1±0.60, 149.9±2.59, 153.4±2.40, 169.1±1.33 and 249±2.06 nm at loading ratios of 0, 65, 130, 195 and 260 µg of capsaicin, respectively (Figure 9). The hydrodynamic diameter is slightly larger in comparison to the effective diameter (Ed); this difference is due to the image analysis of TEM, which reflects the dry state and internal core of the NP. While a dynamic light scattering (DLS) allows measured the NP in solution allowed register the dynamic state of the corona – i.e., the functionalised surface of the NP. The hydrodynamic diameter enables determining the correlation between particle size and physiological process [43]. This overestimation is probably due to the formation of a pseudo-corona composed of some fractions of the long hydrophobic chain with a polar amide group being deposited on the surface of the nanoparticle, increasing the size of the nanoparticles. A similar effect was reported on nanostructured lipid carriers [40]. Besides, at loading ratios of 195 and 260 µg of capsaicin, the super translocation of capsaicin to the surface is the cause of the coalescence between particles and an increase in the size. The mean values of PDI were 0.06±0.03, 0.08±0.02, 0.05±0.03, 0.06±0.00 and 0.14±0.05 at loading ratios of 0, 65, 130, 195 and 260 µg of capsaicin, respectively. The PDI was maintained above 0.4, which indicates that the sample is not polydisperse. On the contrary, the size distribution of the NP is homogeneous between treatments. Probably, the treatment with more formation of aggregates was treatment at 260 µg of capsaicin.
The sentence was changed as follows (Manuscript corrected, Lines 431-488):
The hydrodynamic diameters were 130.1±0.60, 149.9±2.59, 153.4±2.40, 169.1±1.33, and 249±2.06 nm at loading ratios of 0, 65, 130, 195, and 260 µg of capsaicin, respectively (Figure 9). The hydrodynamic diameter is slightly larger than the effective diameter (Ed); this difference is due to the TEM image analysis, which reflects the dry state and internal core of the NPs. Dynamic light scattering (DLS) allows the measurement of NPs in solution, which allows the registration of the dynamic state of the corona, i.e., the functionalised surface of the NPs. The hydrodynamic diameter enables the determination of the correlation between particle size and physiological process [43]. This overestimation is probably due to the formation of a pseudo-corona composed of some fractions of the long hydrophobic chain, with a polar amide group being deposited on the surface of the nanoparticle, thereby increasing the size of the nanoparticles. A similar effect was reported on nanostructured lipid carriers [40]. Additionally, the super-translocation of capsaicin to the surface is the cause of the coalescence between particles and an increase in size at loading ratios of 195 and 260 µg of capsaicin. The mean PDI values were 0.06±0.03, 0.08±0.02, 0.05±0.03, 0.06±0.00, and 0.14±0.05 at loading ratios of 0, 65, 130, 195, and 260 µg of capsaicin, respectively. The PDI was maintained below 0.4, indicating that the sample is not polydisperse; conversely, the distribution of NP size was homogeneous between treatments. The treatment with the highest formation of aggregates was that conducted using 260 µg of capsaicin.
With respect to the observation of Reviewer 2, the reduction of ζ-potential from –34 mV to –24 mV is interpreted as “not compromising stability”. We change this sentence in the abstract section.
Original version (Lines from 39-40):
The ζ-potential decreased due to increase of hydrodynamic diameter; this effect did not compromise the stability of the nanoparticles.
The sentence was changed as follows (Manuscript corrected, Lines 39-40):
The nanoparticles increased in dimensions and showed a loss of shape due to coalescence between nanoparticles.
Moreover, we correct the probability (p=). In the subsection 3.4 ζ-potential, Hydrodynamic Diameter and PDI.
Original version (Line 411):
strong correlation (r=0.97, p=0.05) between the concentration of capsaicin added and the
The sentence was changed as follows (Manuscript corrected, Lines 411):
strong correlation (r=0.97, p=0.0001) between the concentration of capsaicin added and the
Question 4. Polydispersity Index (PDI) interpretation (p.13): The manuscript states that a PDI above 0.4 “indicates that the sample is not polydisperse,” while in fact ISO 22412 and DLS literature classify PDI values >0.4 as indicative of highly polydisperse, heterogeneous samples. This represents a significant conceptual inversion. The text should be corrected to avoid misleading the readers and to comply with standard definitions.
Response: We consider the observation of Reviewer 2 to be valuable, and we sincerely appreciate the contribution to improving the quality of the manuscript. The sentence that mentions Reviewer 2 refers to the reference value to consider a highly polydisperse and heterogeneous sample. The observation of Reviewer 2 is correct. We have confusion, and by mistake during the writing, I wrote “The PDI was maintained above 0.4,” when the correct sentence is “The PDI was maintained below 0.4,”. Because the sentence was corrected.
Original version:
Lines from 445-446:
The PDI was maintained above 0.4, which indicates that the sample is not polydisperse.
The sentence was changed as follows (Manuscript corrected, Lines 445-446):
The PDI was maintained below 0.4, indicating that the sample is not polydisperse;
Question 5. SDS‑PAGE interpretation (p.17): SDS‑PAGE results are described as evidence that the nanoparticles “disassembled” into 66 kDa BSA subunits. However, because SDS-PAGE inherently applies denaturing conditions (SDS, β‑mercaptoethanol, heat), BSA will always appear at 66 kDa, regardless of supramolecular organization. The results confirmed the presence of BSA but did not demonstrate the controlled disassembly of nanoparticles. The interpretation would benefit from rephrasing to avoid overstating what the assay proves.
Response: We consider the observation of Reviewer 2 to be valuable, and we sincerely appreciate the contribution to improving the quality of the manuscript. The section that Reviewer 2 mentioned was “3.7 Stability of the NP”. According to observation of the Reviewer 2 we correct information about the “controlled disassembly of the nanoparticles” and restructure the information of the section to avoid generate confusion in our results.
Original version:
Lines from 551-570:
3.7 Stability of the NP
Although SDS-PAGE is commonly used to determine molecular size and purity of proteins in recent reports this technique was applied to determine the stability of the BSA NP. An efficient crosslinking mechanism protects albumin NP from disassembling in a wide range of environments. However, controlled NP disassembling allows efficient drug release. BSA yielded a broad band in the 66-72 kDa region (Figure 12, lane 2). To demonstrate a crosslinking by heat, a similar quantity of BSA of NP formulations was mixed with capsaicin and dissolved in water at pH 10 without heat crosslinking. The band showed similar molecular weight (MW) and thickness to native BSA (Figure 12, lane 3), which indicates that the albumin does not allow crosslinking. A second control was placed in the gel, a similar formulation to the previous treatment, but with 16 mL of ethanol, without heat crosslinking and the band was not detected because probably albumin was denatured (Figure 12, lane 4). The samples were placed in the gel in ascending order from 0 to 260 µg (Figure 12, lane 5 to 9). The band pattern showed that NP allowed the disassembly in the original units of bovine serum albumin at 66 kDa. Besides, the breadth of the band increased with the concentration of the capsaicin. However, albumin concentration remains constant in all treatments; this effect was probably due to high concentration of capsaicin increasing the propensity of BSA NP to disassemble in SDS-PAGE conditions. These results contrast with Carriles et al. [10], who reported that BSA-capsaicin NP crosslinked with glutaraldehyde did not disassemble even when treated with SDS and β-mercaptoethanol.
The sentence was changed as follows (Manuscript corrected, Lines 552-572):
3.7 SDS‑PAGE of the NP
Although SDS-PAGE is commonly used to determine the molecular size and purity of proteins, this technique has recently been applied in the analysis and comparison of the effect of crosslinking agents in the formulation of BSA NPs [55]. An efficient crosslinking mechanism protects albumin NPs from disassembly in a wide range of environments. BSA yielded a broad band in the 66-72 kDa region (Figure 12, lane 2). To demonstrate heat-based crosslinking, a similar quantity of BSA from NP formulations was mixed with capsaicin and dissolved in water at pH 10 without heat-based crosslinking. The band had a similar molecular weight (MW) and thickness to native BSA (Figure 12, lane 3), which indicates that the albumin does not allow crosslinking. A second control—with a similar formulation to the previous treatment, but with 16 mL of ethanol—was placed in the gel without heat-based crosslinking, and the band was not detected, probably because albumin was denatured (Figure 12, lane 4). The samples were placed in the gel in ascending order of capsaicin concentration from 0 to 260 µg (Figure 12, lanes 5 to 9). The band pattern showed that NPs allowed the disassembly into the original units of bovine serum albumin at 66 kDa. Additionally, the breadth of the band increased with the concentration of capsaicin; however, albumin concentration remained constant across all treatments. This effect was probably due to the high concentration of capsaicin increasing the propensity of BSA NPs to disassemble under SDS-PAGE conditions. These results contrast with those of Carriles et al. [10], who reported that BSA–capsaicin NPs crosslinked with glutaraldehyde did not disassemble, even when treated with SDS, heat, and β-mercaptoethanol.
Reference added
55 Amighi F.; Emam-Djomeh Z.; Labbafi-Mazraeh-Shahi M. Effect of different cross-linking agents on the preparation of bovine serum albumin nanoparticles. J. Iran Chem. Soc. 2020, 17, 1223–1235.
Section 4. Conclusion:
Original version (Lines from 631-632)
The disassembly of the nanoparticles showed efficient capacity to disassemble and deliver the capsaicin.
The sentence was changed as follows (Manuscript corrected, Lines 635-638):
The BSA nanostructured and stabilised with heat crosslinking showed a capacity to denature in BSA at 66 kD. Treatment without heat-based crosslinking of BSA was not detected by SDS-PAGE;
Question 6. Sample size in in vivo studies (p.18–19): The in vivo evaluation used only 12 animals divided into multiple groups. While the findings are suggestive, such a small cohort limits the statistical power and generalizability. I recommend explicitly acknowledging this limitation in the discussion section so that readers can interpret the data with appropriate caution.
Response: We acknowledge that important Reviewer's observation about our animal study. We state its limitations and nature as pilot study in section 3.8, page 26, and at the bottom of the Results/Discussion section, page 28.
Original version:
Lines from 572-575:
3.8 Biocompatibility of BSA-Capsaicin In Vivo
Histopathological analysis of liver tissue sections of treatments at 0, 65 and 260 µg of capsaicin (low and high capsaicin loading ratio) and control (10 mM NaCl) allowed us to determine whether the BSA-capsaicin nanoparticles caused an inflammatory process in the liver.
The sentence was changed as follows (Manuscript corrected, Lines 573-577):
We conducted a pilot study to assess whether the BSA–capsaicin nanoparticles caused an inflammatory reaction in the liver. To this end, liver tissue sections exposed to treatments with 0, 65, and 260 µg of capsaicin (low and high capsaicin loading ratios) and a control (10 mM NaCl) were subjected to histopathological analysis.
The sentence was added as follows (Manuscript corrected, Lines 625-627):
As a caveat, we acknowledge that the sample size of the mice in our study is limited. Although the results are clear, they should be considered preliminary data on which to build larger and more detailed future studies.
Question 7. Generalization in the conclusion (p.20): The conclusion extends the findings with capsaicin to “other hydrophobic drugs.” As no other compounds were tested, this suggestion should be presented with caution. Rephrasing the statement as a hypothesis for future validation, rather than as an established applicability, would improve the accuracy of the conclusions drawn.
Response: In response to Reviewer comment, we accept the suggestion and clarify this point.
Subsection Conclusion:
Original version (Lines from 636-637)
Further work will determine whether this property of capsaicin can be extrapolated to other hydrophobic molecules.
The sentence was changed as follows (Manuscript corrected, Lines 641-644):
These findings support the idea that heat can be used as a crosslinking alternative to glutaraldehyde in the formulation of BSA nanoparticles for encapsulating hydrophobic drugs such as capsaicin.

Round 2
Reviewer 2 Report (Previous Reviewer 1)
Comments and Suggestions for Authors
The authors have addressed the suggestions proposed by this reviewer, and the manuscript has been properly revised and improved.
Author Response
We appreciate the review by Reviewer 2, and we sincerely appreciate the contribution to improving the quality of the manuscript.

This manuscript is a resubmission of an earlier submission. The following is a list of the peer review reports and author responses from that submission.
Round 1
Reviewer 1 Report
Comments and Suggestions for Authors
- Figure 1 shows a linear trend in the encapsulation of capsaicin up to 195 µg/mg of BSA but does not clearly explain the impact of aggregation observed at higher concentrations. Additionally, the figure captions could be more detailed to clarify highlighted points, such as the meaning of the "red circle." How do the authors correlate the formation of aggregates with the practical functionality of nanoparticles?
- Figures 3, 4, 5, 6, and 7 show the morphological changes in the nanoparticles, including increased area, effective diameter, loss of circularity, and changes in the aspect ratio at high capsaicin concentrations. However, there is no robust quantitative analysis of the size distribution or aggregate frequency. How do these morphological changes affect the stability, encapsulation efficiency, biodistribution, and interaction of nanoparticles with cells or tissues?
- Figure 8 shows a decrease in ζ-potential with increasing capsaicin concentration; however, there has been no analysis of how this affects colloidal stability or cell interactions. Long-term stability tests or tests under various environmental conditions, such as light, pH, and humidity, could complement this study. How do the authors ensure that nanoparticles maintain their integrity and functionality in complex biological systems?
- Figures 10 and 11 show the changes in the FTIR and Raman spectra attributed to the interactions between BSA and capsaicin, but there is no direct experimental validation of these interactions. Complementary assays, such as molecular simulations or binding studies, can validate these interactions and explore how they affect encapsulation efficiency and drug release.
- Figure 12 shows the disintegration of nanoparticles in SDS-PAGE, but there was no direct correlation with drug release under physiological conditions. Were performed controlled-release tests of capsaicin. These data are important for connecting the experimental results to the practical functionality of the nanoparticles.
- Although the article mentioned that heat can replace glutaraldehyde as a crosslinking agent, there is no detailed quantitative comparison between the two methods. Is it possible to include comparative data such as encapsulation efficiency, stability, and toxicity to validate the superiority of the proposed method?
- The study concluded that the method can be applied to other hydrophobic compounds; however, there are no experiments to support this generalization. Does the literature suggest this possibility?
- The use of ethanol as a solvent in the synthesis process was not accompanied by the analysis of the solvent residues in the final nanoparticles. Is there a possibility that ethanol residues could compromise the safety or functionality of nanoparticles?
- The choice of 70 °C as the crosslinking temperature was made without comparative analysis with the other temperatures. How do the authors justify this choice and how does it impact the efficiency, stability, and practical applicability of nanoparticles?
- A reduction in nanoparticle yield at high capsaicin concentrations was observed (Figure 2). How do the authors explain this phenomenon and how does it affect the practical applicability of the formulations?
Reviewer 2 Report
Comments and Suggestions for Authors
Please consider the following comments during the revision process.
- Line 108, How can sterile water have a pH of 10?
- The references for the testing protocols must be included.
- Bovine serum albumin melts at 63°C and possibly degrades at temperatures above 65°C. Please justify the stability of BSA at the process temperature (70°C).
- What justification underlies the selection of a process temperature of 70°C? Have any preliminary trials been conducted? If so, kindly include the corresponding data.
- The manuscript must be reviewed for linguistic corrections.
- It is essential to understand the drug release of the developed nanoparticles. Thus, drug/active release studies must be included.
Reviewer 3 Report
Comments and Suggestions for Authors
In the manuscript entitled "Effect of Crosslinking by Heat on the Physicochemical Features of BSA-Capsaicin Nanoparticles ", the authors demonstrated the use of heat as an alternative crosslinking method to glutaraldehyde for the preparation of BSA-capsaicin nanoparticles. While the study offers a thorough materials characterization (including TEM, FTIR, Raman, ζ-potential, etc.), it lacks the biological or pharmacological validation expected for a pharmaceutical journal. The focus lies squarely on material fabrication and structural analysis, which makes it more suitable for a materials science journal rather than Pharmaceutics. Below are detailed comments:
Major Comments:
- µg/mg BSA is not a concentration unit
Throughout the manuscript, the authors use “µg of capsaicin per mg of BSA,” which refers to a mass ratio, not a standard concentration. Please rephrase this as a “loading ratio” or “capsaicin-to-BSA mass ratio” to improve clarity. - Figures 1 and 2 can be merged
Since both represent linear trends (capsaicin loading vs. yield), they can be combined into a single figure or table to reduce redundancy. - The TEM images in Fig. 3d and 3e do not support the aggregation claims made in Figures 4–7
- In Fig. 3d (195 µg/mg), only a few particles are aggregated in pairs, while most appear dispersed or in small multi-particle clusters.
- In Fig. 3e (260 µg/mg), the majority of particles are dispersed, with a small number forming aggregates of three or more—not specifically “four-particle” aggregates.
It is unclear why Figures 4–7 use “two” or “four” particle aggregation states as the basis for analysis. The criteria for these assumptions are unsupported by the TEM data. I recommend reevaluating your aggregation classification and analysis method.
- Figures 4–7 require revision and re-analysis
Since the assumptions about aggregation do not match the actual TEM data, the morphometric analyses derived from these groupings are questionable. Consider an alternative approach, such as classifying particles into “single,” “small aggregates (2–3),” or “large aggregates (≥4)” based on direct counting. - Results and discussion are overly lengthy and poorly organized
The manuscript reads more like a technical report than a scientific article. The discussion lacks structure and is saturated with unprioritized data interpretations, especially in the spectroscopic sections. Please streamline and reorganize the results around key hypotheses and findings. - Lack of biological/pharmaceutical relevance
Despite being submitted to Pharmaceutics, this work does not include any pharmacological or biological assessments such as cytotoxicity, drug release, or therapeutic evaluation. Without such data, it cannot demonstrate pharmaceutical applicability. I strongly recommend the authors consider submitting to a materials-oriented journal such as International Journal of Molecular Sciences (IJMS).
In the manuscript entitled "Effect of Crosslinking by Heat on the Physicochemical Features of BSA-Capsaicin Nanoparticles ", the authors demonstrated the use of heat as an alternative crosslinking method to glutaraldehyde for the preparation of BSA-capsaicin nanoparticles. While the study offers a thorough materials characterization (including TEM, FTIR, Raman, ζ-potential, etc.), it lacks the biological or pharmacological validation expected for a pharmaceutical journal. The focus lies squarely on material fabrication and structural analysis, which makes it more suitable for a materials science journal rather than Pharmaceutics. Below are detailed comments:
Major Comments:
- µg/mg BSA is not a concentration unit
Throughout the manuscript, the authors use “µg of capsaicin per mg of BSA,” which refers to a mass ratio, not a standard concentration. Please rephrase this as a “loading ratio” or “capsaicin-to-BSA mass ratio” to improve clarity. - Figures 1 and 2 can be merged
Since both represent linear trends (capsaicin loading vs. yield), they can be combined into a single figure or table to reduce redundancy. - The TEM images in Fig. 3d and 3e do not support the aggregation claims made in Figures 4–7
- In Fig. 3d (195 µg/mg), only a few particles are aggregated in pairs, while most appear dispersed or in small multi-particle clusters.
- In Fig. 3e (260 µg/mg), the majority of particles are dispersed, with a small number forming aggregates of three or more—not specifically “four-particle” aggregates.
It is unclear why Figures 4–7 use “two” or “four” particle aggregation states as the basis for analysis. The criteria for these assumptions are unsupported by the TEM data. I recommend reevaluating your aggregation classification and analysis method.
- Figures 4–7 require revision and re-analysis
Since the assumptions about aggregation do not match the actual TEM data, the morphometric analyses derived from these groupings are questionable. Consider an alternative approach, such as classifying particles into “single,” “small aggregates (2–3),” or “large aggregates (≥4)” based on direct counting. - Results and discussion are overly lengthy and poorly organized
The manuscript reads more like a technical report than a scientific article. The discussion lacks structure and is saturated with unprioritized data interpretations, especially in the spectroscopic sections. Please streamline and reorganize the results around key hypotheses and findings. - Lack of biological/pharmaceutical relevance
Despite being submitted to Pharmaceutics, this work does not include any pharmacological or biological assessments such as cytotoxicity, drug release, or therapeutic evaluation. Without such data, it cannot demonstrate pharmaceutical applicability. I strongly recommend the authors consider submitting to a materials-oriented journal such as International Journal of Molecular Sciences (IJMS).